# FedSMU: Communication-Efficient and Generalization-Enhanced Federated Learning through Symbolic Model Updates

## Abstract

The significant communication overhead and client data heterogeneity have posed important challenges to current federated learning (FL) paradigm. Most compression-based and optimization-based FL algorithms typically focus on addressing either the model compression challenge or the data heterogeneity issue individually, rather than tackling both of them. In this paper, we observe that by symbolizing the client model updates to be uploaded (i.e., normalizing the magnitude for each model parameter at local clients), the model heterogeneity can be mitigated that is essentially stemmed from data heterogeneity, thereby helping improve the overall generalization performance of the globally aggregated model at the server. Inspired with this observation, and further motivated by the success of Lion optimizer in achieving the optimal performance on most tasks in centralized learning, we propose a new FL algorithm, called FedSMU, which simultaneously reduces the communication overhead and alleviates the data heterogeneity issue. Specifically, FedSMU splits the standard Lion optimizer into the local updates and global execution, where only the symbol of client model updates commutes between the client and server. We theoretically prove the convergence of FedSMU for the general non-convex settings. Through extensive experimental evaluations on several benchmark datasets, we demonstrate that our FedSMU algorithm not only reduces the communication overhead, but also achieves a better generalization performance than the other compression-based and optimization-based baselines.

## 1 Introduction

Federated learning (FL) is a large-scale machine learning paradigm wherein a multitude of clients, under the orchestration of a central server, collaboratively learn a model without the need of sharing or exchanging any raw client data (McMahan et al., 2017). This paradigm is commonly adopted in data-constrained or data-sensitive environments, such as Internet of things (IoT), healthcare, and finance (Khan et al., 2021; Rieke et al., 2020; Yang et al., 2019). In essence, FL is distinguished from the traditional distributed learning in the following three major challenges.

- **High communication cost.** During each communication round of training, the clients are required to transmit their local model parameters (or updates) to the central server for the global aggregation. When the number of model parameters becomes significantly large, this transmission process may result in a huge bandwidth consumption.

- **Data heterogeneity.** Due to the inherently private and personalized nature of federated clients, the datasets across these clients tend to exhibit distinct statistical distributions. Such a data heterogeneity may introduce significant biases into the globally learned model, which, in turn, can diminish the global model's generalization capability.

- **Partial client participation.** In practical scenarios, clients may join or leave the FL system at random time intervals. This highly dynamic behavior results in only a small subset of clients being active for training during each communication round.

To address these challenges, extensive exploration has been conducted in the FL community, but from different perspectives. On one hand, compression-based federated algorithms aim to reduce the amount of data required for model parameter (or update) transmission, including both the unbiased

compression and biased compression. For instance, unbiased compression, such as SignSGD (Bernstein et al., 2018a;b), QSGD (Alistarh et al., 2017) and FedPAQ (Reisizadeh et al., 2020), quantize the gradient values into lower-precision integers, thereby reducing the number of transmitted bits. While the common biased compression approach is to sparsify the gradient vector by setting some of its elements to zero or very small values, with the aim of reducing the data transmission cost (Wangni et al., 2018; Aji & Heafield, 2017; Lin et al., 2017). However, the direct use of compression methods will lose a certain amount of information, resulting in the problem of decrease in the accuracy (Yu et al., 2022) and slower convergence or even divergence (Beznosikov et al., 2023). Some methods, such as Error Feedback (Richtárik et al., 2021), have been designed to mitigate these issues by incorporating error feedback into the optimization process.

Several optimization-based federated algorithms, on the other hand, have been proposed to address the data heterogeneity issue. For example, SCAFFOLD (Karimireddy et al., 2020) aims to mitigate the client variance by designing and iteratively updating the control variates. Though theoretically effective, it incurs doubling the communication overhead. FedGen (Venkateswaran et al., 2023) regulates the local training by transmitting additional generators. Most of these optimization-based FL algorithms, which mainly aim at mitigating data heterogeneity, may incur additional communication overhead of information exchange for performance improvement. Additionally, it remains unknown whether these optimization-based algorithms are compatible with the current compression techniques.

In this paper, we aim to design an algorithm capable of simultaneously addressing the communication bottleneck and data heterogeneity, without being constrained by the partial client participation issue. To achieve this goal, we first revisit the typical FedAvg algorithm and identify that heterogeneous magnitudes of model updates may result in certain clients' updates being overlooked, thus leading to an unstable and sub-optimal aggregation of the global model. Building upon this observation, we then introduce the concept of "Magnitude Uniformity" index, which quantifies the clients' contribution to the global model's update. We empirically validate that this magnitude uniformity index is influenced by the degree of data heterogeneity in federated learning, indicating that a more heterogeneous data distribution leads to a greater heterogeneity in the magnitudes of client model updates. Furthermore, heterogeneous client updates may contribute to a decline in the global model's generalization performance. To address this issue and further reduce the communication overhead, we are motivated to symbolize the model updates as an immediate solution, and propose the FedSMU algorithm. Our contributions can be summarized as follows.

- We develop a compression-based FL algorithm called FedSMU. It uses the sign operation to achieve a 1-bit compression and thus greatly saves the communication cost. Simultaneously, we leverage the design of Lion optimizer(Chen et al., 2024) to enhance the generalization performance while maintaining the benefits of compression.

- We conduct a convergence analysis of FedSMU under the general non-convex settings, and find its convergence rate as $\mathcal{O}(\frac{1}{\sqrt{T}})$, where $T$ is the total number of communication rounds. This theoretical result matches with the convergence rates of existing FL algorithms.

- We conduct a series of experiments to demonstrate the superiority of FedSMU. By comparing FedSMU with the other compression-based and optimization-based federated algorithms, we show that our FedSMU algorithm achieves a higher generalization performance while greatly saving the communication overhead at most cases.

## 2 RELATED WORKS

**Compression-Efficient FL.** Extensive studies have been dedicated to reducing the amount of data required for gradient transmission and thus improving the communication efficiency. Using the method called unbiased compression, QSGD (Alistarh et al., 2017), FedPAQ (Reisizadeh et al., 2020), ZipML (Zhang et al., 2017) and ECQ-SGD (Wu et al., 2018) compress the gradients uploaded to the server while keeping the original data integrity and expectation unchanged to save the communication cost. By leveraging the sign operation, signSGD (Bernstein et al., 2018a;b) and TernGrad (Wen et al., 2017) can compress the gradients up to 1 bit. While the sparsification-based method like TopK (Stich et al., 2018; Alistarh et al., 2018), which only keeps the largest $K$ gradients, is another communication-efficiently biased compression method. Some other methods, like FedEF (Li & Li, 2023), FedZip (Malekijoo et al., 2021) and Qsparse-local-SGD (Basu et al., 2019), incorporate both the quantization and sparification. While a direct application of biased compression on gradient into

federated learning, such as Top-k, can lead to a performance degradation and lower convergence speed due to bias accumulation (Beznosikov et al., 2023). To address this, some studies have introduced optimization techniques to mitigate the negative effects of bias. For example, EF21 (Richtárik et al., 2021) employs the error feedback, while MARINA (Gorbunov et al., 2021) and DIANA (Mishchenko et al., 2024) leverage the compression of gradient differences, both of which further enhance the model performance and convergence speed. In this work, we adopt the sign operation to achieve the communication efficiency, which also helps enhance the generalization capability of the globally aggregated model as shown by Chen et al. (2024; 2021); Foret et al. (2020).

**Generalization-Enhanced FL.** In the advancement of FL algorithms, various techniques have emerged to improve the generalization performance. By using momentum in FL, one can track the historical information of gradients, suppress the noise and reduce the instability of model updates. Benefiting from this, methods such as MV-sto-signSGD-SIM (Sun et al., 2023) and FedAdam (Reddi et al., 2020) apply momentum instead of directly updating with gradients, while PR-SGD-Momentum (Yu et al., 2019) first updates the momentum and then combines the new gradient with a weight of the momentum. These methods enhance model generalization and accelerate convergence in FL. In this work, we employ two sliding average functions to update momentum after calculating the new gradient, a technique demonstrated by Lion (Chen et al., 2024) to effectively store more historical gradient data. Also, since weight decay regularization has been shown to outperform $\ell_2$ regularization in preventing overfitting and enhancing generalization (Loshchilov, 2017), we leverage a weight decay strategy to mitigate the impact of data heterogeneity and further improve generalization performance. Distributed Lion (Liu et al., 2024) is a distributed learning algorithm that leverages the Lion optimizer (Chen et al., 2024) focusing on reducing communication overhead. It extends the Lion optimizer to the distributed setting with full client participation and IID data. However, it lacks exploration of partial participation and non-IID data scenarios, which are the major challenges in FL.

## 3 PROPOSED METHOD

### 3.1 NOTATIONS AND PRELIMINARIES

The general optimization problem of federated learning (FL) can be formulated as:

$$\min_{x \in \mathbb{R}^d} f(x) := \frac{1}{m} \sum_{i=1}^{m} F_i(x), \quad (1)$$

where $F_i(x) \triangleq \mathbb{E}_{\xi \sim D_i}[F_i(x, \xi)]$ represents the local loss function of the $i$-th client with the data sample $\xi$ drawn from distribution $D_i$. Under the FL settings, data is typically heterogeneous, implying

Table 1: Summary of notations.

| | |
|---|---|
| $T, t$ | number, index of communication rounds |
| $K, k$ | number, index of local update step |
| $\eta, \gamma_1$ | local, global learning rate |
| $y_{t,k}^i$ | client $i$'s model at round $t$ and step $k$ |
| $x_t$ | aggregated server model after round $t$ |
| $\mathcal{M}, m$ | set of clients with cardinality $m$ |
| $\mathcal{N}_t, n$ | set of sampled active clients with cardinality $n$ |

that for different clients $i$ and $j$, the distributions $D_i$ and $D_j$ can be extremely different. Moreover, the FL systems often operate under a limited bandwidth, which renders the communication overhead associated with the exchange of model parameters a significant bottleneck.

Current approaches in FL often prioritize either mitigating data heterogeneity to enhance generalization or compressing model updates to alleviate communication, rather than addressing both challenges concurrently. Specifically, most compression-based FL algorithms (Bernstein et al., 2018a;b; Li & Li, 2023; Wen et al., 2017) significantly reduce communication costs, with generalization performance typically comparable to or slightly lower than that of standard FedAvg (McMahan et al., 2017). On the other hand, most optimization-based FL strategies (Karimireddy et al., 2020), which involve exchange of full-precision model updates, and even additional control variables or informative representations, aim to mitigate the data heterogeneity issue, but at the cost of a huge communication overhead.

The recently proposed SCALLION algorithm (Huang et al., 2023) integrates the control variable-based SCAFFOLD framework with incremental variable compression methods, achieving a comparable performance with SCAFFOLD while substantially reducing the upload communication cost. Nonetheless, SCALLION additionally requires to double the download communication overhead for the transmission of control variables. This observation then imposes a critical question for the field of compression-efficient FL: can we design an approach that effectively mitigates both the communication bottleneck and data heterogeneity simultaneously?

## 3.2 Symbolizing Client Updates

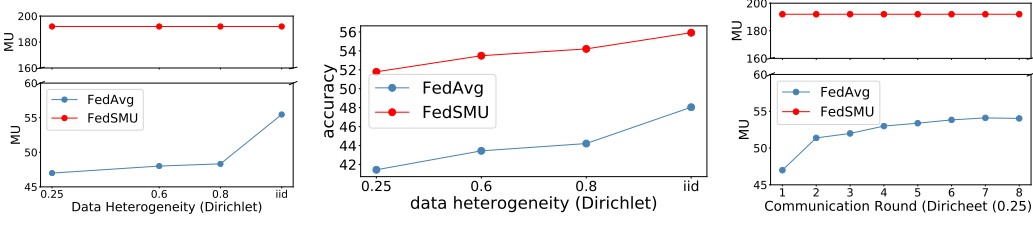

(a) MU index vs. heterogeneity  (b) Accuracy vs. heterogeneity  (c) MU vs. communication round

Figure 1: Magnitude uniformity (MU) index and top validation accuracy of FedAvg and FedSMU (ours) on CIFAR-100 with small CNN network.

Before answering this question, we revisit the standard FL framework, i.e., FedAvg (McMahan et al., 2017). With FedAvg, clients perform local training using their own datasets that are distributed over clients and non-iid in nature. The server then aggregates these locally trained models to update the global model, which subsequently serves as the initial model for the next round of training. However, due to data heterogeneity, clients' model updates often differ in both direction and magnitude. Consequently, when model updates from different clients with large deviations are averaged, some updates with relatively small magnitudes may be overlooked. For instance, we consider three clients, $i_1$, $i_2$ and $i_3$, whose model updates along one dimension are $+10$, $-1$, and $-1$, respectively. In this case, the updates have opposite directions, while the magnitude of client $i_1$'s update is much larger than those of clients $i_2$ and $i_3$. After averaging at the server (i.e., the global model's update becoming $+8/3$), the contribution of clients $i_2$ and $i_3$ to the global model's update will be ignored, since the update direction is now only dominantly determined by client $i_1$. Therefore, a direct averaging may neglect contributions from the smaller updates and potentially compromise the fairness among clients.

To address this issue, and motivated by the Jain's fairness index (Jain et al., 1984), we propose a new metric called the Magnitude Uniformity index to reflect the clients' contribution to the global model update. Through empirical analysis, we explore the relationship between this Magnitude Uniformity index and the local data heterogeneity, which in turn impacts the generalization performance of the globally aggregated model.

**Definition 3.1.** (**Magnitude Uniformity**). We define the magnitude uniformity across $m$ clients at the communication round $t$ as:

$$\Phi_t \triangleq \sum_{j=1}^{d} \frac{\left(\sum_{i \in \mathcal{M}} \hat{g}_t^{i,j}\right)^2}{\|\mathcal{M}\| \sum_{i \in \mathcal{M}} \left(\hat{g}_t^{i,j}\right)^2}, \ \hat{g}_t^{i,j} = \|y_{t,K}^{i,j} - y_{t,0}^{i,j}\|, \tag{2}$$

where $y_{t,K}^{i,j}$ denotes the $j$-th dimension of client $i$'s model at round $t$ and local step $K$, and $\hat{g}_t^{i,j}$ denotes the magnitude of client $i$'s model update in this dimension $j$ at round $t$. Similar to the Jain's fairness index, a higher value of the magnitude uniformity $\Phi_t$ indicates a more uniform contribution from the clients, thus suggesting a more balanced representation of the clients' data in the global model. Theoretically, such a uniformity may lead to a global model that better captures the information from all the local clients. Consequently, one might raise the following question: does this magnitude uniformity index get affected by the data heterogeneity across locally distributed clients, and does it further influence the global model's generalization performance?

Seeking for the answer to this question, we empirically examine the correlation between this Magnitude Uniformity index and the global model's generalization performance under varying data heterogeneity with the CIFAR-100 dataset. The experiment involves 100 clients with a participation rate of 10%. As observed from Figures 1(a) and 1(b), for FedAvg, an increase in data heterogeneity leads to a decrease in the Magnitude Uniformity index, accompanied by a deterioration in the generalization performance. This suggests that with FedAvg, data heterogeneity leads to a significant difference in the magnitude of model updates across clients, resulting in an unstable global aggregation and poorer generalization performance. Additionally, as shown in Figure 1(c), the Magnitude Uniformity index tends to rise during the FedAvg training, suggesting that the early stage of an FL system forces a gradual narrowing on the magnitude difference of model updates across clients.

A straightforward approach to enhance the Magnitude Uniformity index for FL is to apply a sign operation to the local clients' updates, ensuring that model updates have the uniform magnitude

from all the clients. Specifically, after this sign operation, the local model updates for the three clients $i_1$, $i_2$ and $i_3$ in the previous example would become $+1$, $-1$, and $-1$, respectively. This process guarantees that each client's model update contributes equally to the globally aggregated model, thereby reducing the impact of model heterogeneity and promoting fairness. By converting the magnitudes of model updates into their respective signs, we actually emphasize the directions of their updates rather than their magnitudes, which could help balance the contributions from different clients' model updates and lead to a more representative and informative global model. Moreover, by symbolizing the updates we can reduce the communication cost to 1 bit per dimension, offering a potential solution to enhancing generalization while also saving the communication.

In fact, numerous sign-based compression methods (Bernstein et al., 2018a;b; Wen et al., 2017) have been applied in federated learning. While theoretically performant, their empirical results often show only marginal improvements or comparable performance to FedAvg. Thus, effectively leveraging the sign operation to simultaneously mitigate the communication overhead and enhance generalization in federated learning remains a challenging and unresolved issue. On the other hand, many optimization techniques have been proposed to improve generalization for the centralized learning, such as momentum, Adam, and weight decay. A brute force approach could be directly incorporating the sign operations with these optimization strategies in FL, formulating the algorithm design as a program search to identify federated optimization algorithms that can incorporate sign compression. However, this approach is computationally expensive.

Fortunately, in the context of centralized learning, Lion (EvoLved Sign Momentum) optimizer (Chen et al., 2024) employs the sign operation to compute the updates while tracking momentum. This approach has demonstrated an overall outstanding performance across a variety of models and tasks. Compared to the simple SignSGD (Bernstein et al., 2018a;b), Lion leverages the dual momentum tracking and weight decay, significantly improving the generalization ability of the trained models. Inspired with our observation on the impact of Magnitude Uniformity index on the FL algorithm's generalization performance, and further motivated by the success of Lion in centralized learning, we thus propose a new federated optimization algorithm aiming at both reducing the communication overhead and enhancing generalization performance, through symbolizing the client model updates.

### 3.3 PROPOSED FEDSMU

To leverage the structured design of Lion optimizer and minimize the communication overhead, we propose our FedSMU algorithm for federated learning, which splits the Lion optimizer's framework of momentum tracking and weight decay to be carried out independently at the server and each client, respectively, as summarized in Algorithm 1.

Specifically, at each communication round $t$, FedSMU implements the following steps:

1. Participating clients initialize their local models, denoted as $y_{t,1}^i$, based on the current global model $x_t$.

2. Each client conducts $K$ steps of local stochastic gradient descent (SGD) to compute the model update $g_t^i$.

3. Each client symbolically represents its model updates using momentum and sign operations.

4. The server receives and aggregates these symbolic updates, denoted as $u_t^i$, to update the global model $x_{t+1}$ by incorporating the weight decay.

Such a design offers two significant advantages for our FedSMU algorithm. First, it fully leverages the structure of the Lion optimizer, thereby enhancing the generalization performance of the global model. It is also worth noting that in scenarios where the number of local update steps is set to $K = 1$, our optimizer essentially reverts to the standard Lion. Second, by transmitting only 1-bit update for each dimension of the model parameters between the clients and server, we substantially reduce the communication overhead in the FL systems.

We also notice that inspired by the advantages of the Lion optimizer in centralized learning, there has been other works (e.g., FedLion (Tang & Chang, 2024)) incorporating Lion into the local updates of federated learning. However, FedLion simply uses the vanilla Lion algorithm for the local updates instead of SGD, resulting in a communication cost that is even significantly higher than those of FedAvg, as the extra momentum terms need to be transmitted. Compared with FedLion, our FedSMU out-stands in the following two advantages. **1) Effective utilization of the Lion framework.** Our

---

**Algorithm 1:** Federated learning through Symbolic Model Updates (FedSMU) algorithm.

---

1 **Server Initialization**: $x_0$;
2 **Client Initialization**: $m_0(i) = 0$;
3 **for** *each round* $t = 1, 2, \dots T$ **do**
4     sample clients $\mathcal{N}_t \subseteq \mathcal{M}$
5     **for** *each client* $i \in \mathcal{N}_t$ *in parallel* **do**
6        receive and initialize local model $y_{t,0}^i = x_t$
7        **for** *each local step* $k = 1, 2, \dots, K$ **do**
8           $y_{t,k}^i = y_{t,k-1}^i - \eta \nabla F_i(y_{t,k-1}^i, \xi_{t,k-1}^i)$
9        **end**
10        $g_t^i = y_{t,K}^i - y_{t,0}^i$
11        $u_t^i = \mathbf{sign}(\beta_1 m_{t-1}^i + (1 - \beta_1) g_t^i)$
12        $m_t^i = \beta_2 m_{t-1}^i + (1 - \beta_2) g_t^i$ (for $i \notin \mathcal{N}_t, m_t^i = m_{t-1}^i$)
13        send $u_t^i$ to server
14     **end**
15     // at server:
16     $x_{t+1} = x_t + \gamma_1(\frac{1}{n}\sum_{i=1}^n u_t^i - \gamma_2 x_t)$
17     broadcast $x_{t+1}$
18 **end**

---

FedSMU divides the execution of Lion optimizer between the clients and server. Specifically, when the number of local update steps and the number of clients are set to $K = 1$ and $m = 1$, respectively, the entire federated learning process reduces to the standard Lion algorithm. In contrast, FedLion merely executes the Lion algorithm locally in parallel as a local optimization strategy, failing to exploit the complete structure of Lion. **2) Communication overhead saving.** In addition to the model updates, FedLion requires the additional transmission of the full-precision momentum terms, resulting in a significantly higher communication cost compared to our FedSMU that only necessitates a 1-bit communication for each dimension of the model updates. This substantial reduction in communication overhead is another key advantage of our FedSMU.

## 4 THEORETICAL RESULTS ON CONVERGENCE

We now present the convergence analysis of our proposed FedSMU for the general non-convex functions. In general, our analysis is based on the following three standard assumptions, which are commonly satisfied by a range of non-convex objective functions.

**Assumption 4.1.** (Lipschitz Gradient). *For all $i \in \mathcal{M}$, the function $F_i$ is L-smooth:* $||\nabla F_i(\boldsymbol{x}) - \nabla F_i(\boldsymbol{y})|| \leq L||\boldsymbol{x} - \boldsymbol{y}||$ *for all $\boldsymbol{x}, \boldsymbol{y} \in \mathbb{R}^d$.*

**Assumption 4.2.** (Bounded Variance). *For all $i \in \mathcal{M}$, the function $F_i$ have local-bounded variance $\sigma_l^2$:* $\mathbb{E}[||\nabla F_i(\boldsymbol{x}, \xi) - \nabla F_i(\boldsymbol{x})||]^2 \leq \sigma_l^2$ *for all $\boldsymbol{x} \in \mathbb{R}^d$.*

**Assumption 4.3.** (Bounded Gradients). *For all $i \in \mathcal{M}$, the function $F_i(\boldsymbol{x}, \xi)$ have G-bounded gradient:* $||\nabla F_i(\boldsymbol{x}, \xi)|| \leq G$ *for all $\boldsymbol{x} \in \mathbb{R}^d$.*

For the non-convex optimization problem, Assumptions 4.1 and 4.2 are standard and widely adopted in various literature of FL (Reddi et al., 2020; Bottou et al., 2018; Reddi et al., 2016; Ghadimi & Lan, 2013; Li & Orabona, 2019). Assumption 4.3 is commonly used in convergence analysis of sign-based method like distributed signSGD (Sun et al., 2023; Jin et al., 2020a).

**Theorem 4.4.** *Under Assumptions 4.1, 4.2, and 4.3, when $0 < \eta \leq \frac{1}{4LK}$, $\gamma_1 = \mathcal{O}(\frac{1}{L\sqrt{T}})$ and $1 - \beta_1 = \mathcal{O}(\frac{1}{\sqrt{T}})$, we have:*

$$\Psi \leq \frac{L(f(\boldsymbol{x}_0) - \min f)}{\sqrt{T}} + \frac{3\sqrt{d}G\phi}{nT(1-\beta_2)} + \frac{6d\tau_{max}}{(1-\beta_2)\sqrt{T}} + \frac{3\sqrt{d}G\eta(1+\beta_2)}{(1-\beta_2)L\sqrt{T}} \tag{3}$$

$$+ \frac{3\sqrt{d}G}{(1-\beta_2)\sqrt{T}} + \frac{6\sqrt{d}\eta}{\sqrt{T}}\sqrt{\frac{1-\beta_2}{1+\beta_2}(2\sigma_l^2 + 4K\sigma_l^2 + 4KG^2)} + \frac{6dG}{L\sqrt{T}} + \frac{2d}{\sqrt{T}},$$

*where* $\Psi = \frac{1}{T}\sum_{t=1}^{T}\mathbb{E}[||\nabla f(\boldsymbol{x}_t)||_1]$, $\phi = \sum_{i=1}^{m}\|\frac{1}{G}\nabla F_i(\boldsymbol{x}_0)\|$, $d$ *denotes the dimensions of parameters,* $\tau_{max} = \max\{\tau^i\}_{1\leq i\leq m, 1\leq t\leq T}$ *and* $\tau^i$ *denotes cilent* $i$*'s participation interval. Note that in cases of non-uniform random participation or varying participation rates across rounds,* $\mathbb{E}(\tau_{max})$ *may not equal to* $\frac{m}{n}$.

*Proof.* See Appendix D for the detailed proof. □

*Remark* 4.5. The convergence rate of our FedSMU is $\mathcal{O}(\frac{1}{\sqrt{T}})$ when $T$ is sufficiently large, matching with the convergence rates of existing FL algorithms, such as FedAvg and FedPAQ (Reisizadeh et al., 2020). Note that $\tau_{max}$ represents the maximum participation interval among all the clients, indicating that larger participation intervals result in a slower convergence. Note that $d$ represents the model dimension and directly influences the rate of convergence, i.e., a larger model dimension results in slower convergence. In this analysis, we use a 1-bit quantization compression method. If a higher-bit compression (e.g., $\alpha$-bit) is used, the additional coefficient $\alpha$ will further slow down the overall convergence rate.

*Remark* 4.6. In addition, the original work of Lion does not include the convergence analysis. Our theoretical analysis also provides the relevant convergence rate for the Lion optimizer. Specifically, by setting $n = 1, \tau_{max} = 1$ and $K = 1$, the convergence rate of the our FedSMU will reduce to that of the Lion optimizer.

## 5 EXPERIMENTS

We conduct comprehensive comparative experiments to validate the superior performance of FedSMU in scenarios involving different partial participation rates and data heterogeneity degrees. Further, a series of ablation experiments are designed to corroborate the effectiveness and necessity of FedSMU. The code is available at https://anonymous.4open.science/r/fedsmu-400D.

### 5.1 EXPERIMENTAL SETUP

**Models and Dataset.** We evaluate FedSMU and the other baseline algorithms on three real-world visual and language datasets: CIFAR-10, CIFAR-100 (Krizhevsky et al., 2009) and neural machine translation on Shakespeare, with the same train/test splits as in (Acar et al., 2021). Each client is assigned an uncertain number of classes, and the data within each class varies widely, with the labels of client samples generated according to the Dirichlet distribution. For instance, using Dirichlet-0.25 on CIFAR-10, there are approximately 80% of each client's samples belonging to around three or four different classes. We employ CNN and RNN models similar to previous studies (McMahan et al., 2017). Furthermore, to demonstrate the applicability of our approach to other models, we also evaluate the performance of our algorithms using a larger network, ResNet18. For additional details on the experimental setup, please refer to Appendix A.

**Comparison Algorithms.** We compare the validation (test) performance of our FedSMU with several other baselines, including the optimization-based FL algorithms such as FedAvg (McMahan et al., 2017), FedLion (Tang & Chang, 2024), and SCAFFOLD (Karimireddy et al., 2020), as well as the compression-based FL algorithms such as FedEF-SGD-hv-sign (Li & Li, 2023), FedEF-SGD-topk (Li & Li, 2023), FedEF-SGD-sign (Li & Li, 2023), and SCALLION (Huang et al., 2023). It is worth noting that FedLion (Tang & Chang, 2024) involves a parallel execution of the Lion optimizer on the local clients, requiring the upload of full-precision momentum updates in addition to the compressed model updates. Consequently, the communication overhead of FedLion is higher than FedAvg, even when the model updates are compressed. Additionally, SCAFFOLD needs twice of the communication cost compared to FedAvg. Though SCALLION uploads the compressed incremental updates, it still results in doubling the communication overhead during the download phase.

**Implementation.** We evaluate the performance of the global model after 4000 communication rounds on the CIFAR-10 and CIFAR-100 datasets, utilizing 100 clients with high (H) and low (L) client participation rates of 10% and 3%, respectively. For the ResNet18 (with 10 clients) and RNN (with 100 clients), we adopt the client participation rate of 30% and 10%, respectively. Clients are uniformly sampled at random without replacement at each round. The learning rates and hyperparameters for all approaches are individually tuned via a grid search. For additional details on hyperparameter settings, please refer to Appendix A.

Table 2: Performance comparison under various settings, where a smaller Dirichlet parameter indicates a higher data heterogeneity, and L and H indicate low and high participation rates, respectively. For CIFAR-10 and CIFAR-100, a small CNN network is used, and for Shakespeare, an RNN network is employed. Bold numbers indicate the best performance.

| Dataset | Setting | FedAvg | SCAFFOLD | SCALLION | FedEF-HS | FedEF-TopK | FedEF-Sign | FedLion | FedSMU |
|---|---|---|---|---|---|---|---|---|---|
| | | | | Top-1 Test Accuracy (%). | | | | | |
| CIFAR-100 | Dir (0.25)-L | 41.44 | 41.28 | 42.68 | 38.31 | 44.29 | 37.79 | 45.09 | **49.8** |
| | Dir (0.6)-L | 41.36 | 45.04 | 43.28 | 38.63 | 44.41 | 40.34 | 47.19 | **51.65** |
| | Dir (0.25)-H | 42.29 | 50.49 | 45.24 | 37.03 | 42.69 | 36.12 | 48.33 | **51.79** |
| | Dir (0.6)-H | 43.44 | 50.02 | 45.84 | 36.09 | 42.72 | 38.99 | 48.85 | **53.49** |
| CIFAR-10 | Dir (0.25)-L | 80.95 | **81.6** | 80.91 | 78.35 | 80.11 | 77.87 | 79.04 | 79.66 |
| | Dir (0.6)-L | 82.42 | 82.36 | 81.18 | 79.29 | 81.73 | 79.68 | 80.94 | 81.96 |
| | Dir (0.25)-H | 80.6 | **83.31** | 81.42 | 78.17 | 79.92 | 78.34 | 81.61 | 80.67 |
| | Dir (0.6)-H | 81.43 | **84.12** | 81.75 | 78.75 | 81.42 | 79.38 | 83.15 | 82.66 |
| Shakespeare | noniid-H | 47.58 | **51.28** | 47.86 | 45.79 | 46.21 | 45 | 47.11 | 47.81 |

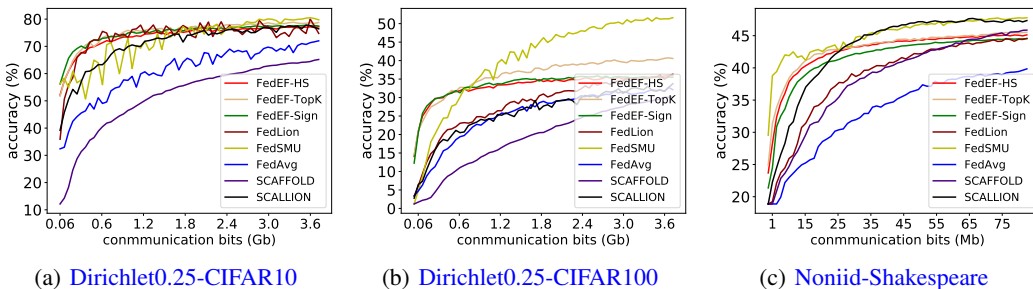

(a) Dirichlet0.25-CIFAR10   (b) Dirichlet0.25-CIFAR100   (c) Noniid-Shakespeare

Figure 2: Convergence performance vs. number of communication bits on CIFAR-10, CIFAR-100 and Shakespeare, with 100 clients and 10% participation for different algorithms. For CIFAR-10 and CIFAR-100, a small CNN network is used, and for Shakespeare, an RNN network is employed.

## 5.2 EXPERIMENTAL RESULTS

### 5.2.1 PERFORMANCE EVALUATION

Experimental results for all the comparison methods under three datasets are shown in Table 2 and Figure 2. In most cases, our FedSMU demonstrates a superior performance compared to the other baselines (especially compression-based) with varying data distributions and client participation rates. The results effectively demonstrate that our algorithm performs well on both image classification and text prediction tasks. We attribute this improvement to our design, which mimics the Lion optimizer and incorporates symbolic updates, momentum tracking, and weight decay. In contrast, other compression methods, such as TopK and group sign employed by FedEF-TopK and FedEF-Sign, compress the communication traffic but consistently exhibit a poorer generalization performance.

Note that our FedSMU generally presents a more significant performance gain on CIFAR-100 for image classification. For CIFAR-10, though our FedSMU outperforms the compression-based FL algorithms, it is still less effective than the optimization-based algorithms, such as FedAvg and SCAFFOLD. Here, we discuss about the possible reason for this slight degradation on CIFAR-10. In a federated heterogeneous scenario involving CIFAR-100, which comprises 100 categories as compared to 10 categories for CIFAR-10, each client typically handles a subset of 13-16 (or 20-25) categories when setting $Dir = 0.25$ (or $Dir = 0.6$). Consequently, with such a high degree of heterogeneity incurred in CIFAR-100, the model updates from clients are more deviated, allowing our FedSMU to be more effective and demonstrate a more significant improvement than on CIFAR-10.

To confirm that our algorithm can maintain a good performance in larger network models, we also conduct comparative experiments on the ResNet18. The experimental results are shown in Table 3. The results indicate that FedSMU demonstrates strong performance on ResNet18 for both CIFAR-10 and CIFAR-100, outperforming most baseline methods, though it remains slightly below SCAFFOLD on the CIFAR-10 dataset.

Table 3: Performance comparison under various settings with the ResNet18 network model. Bold numbers indicate the best performance.

| Dataset | Setting | FedAvg | SCAFFOLD | SCALLION | FedEF-HS | FedEF-TopK | FedLion | FedSMU |
|---|---|---|---|---|---|---|---|---|
| | | | Top-1 Test Accuracy (%). | | | | | |
| CIFAR-10 | Dir (0.25) | 81.74 | **85.62** | 83.75 | 80.43 | 82.54 | 82.44 | 83.28 |
| CIFAR-100 | Di r(0.25) | 47.41 | 48.76 | 48.15 | 47.24 | 48.07 | 43.75 | **49.00** |

### 5.2.2 GENERALIZATION VS. PARTICIPATION RATE

We then evaluate the effect of different participation rates on all the algorithms, while keeping the number of participating clients consistent at each communication round. Results in Table 4 indicate that FedSMU achieves the highest accuracy on most cases. Specifically, when the number of participating clients is maintained at 10, and when the participation rate decreases from 0.2 to 0.05, FedAvg (McMahan et al., 2017), FedEF-SGD-TopK, FedEF-SGD-sign (Li & Li, 2023), SCAFFOLD (Karimireddy et al., 2020), and SCALLION (Huang et al., 2023) would experience a severe performance deterioration of 7.18%, 9.84%, 11.64%, 15.21%, and 12.58%, respectively. In contrast, FedSMU maintains a more stable and superior performance, with only a 1.97% deterioration. These results indicate that our algorithm is minimally impacted by client participation rates and demonstrates greater stability under partial client participation. We would like to attribute this to the use of symbolic operations for client updates, which effectively leverages each client's update even at very low participation rates. Specifically, when the client participation rate is low, data heterogeneity may cause the update of certain clients to dominate due to larger magnitudes. Symbolic operations can mitigate this by normalizing the update amplitudes, ensuring that the contributions of all clients are fully considered. However, whether other structures of the Lion optimizer also play a significant role in mitigating the influence of data heterogeneity remains an open question and an interesting direction for future research.

Table 4: Top validation accuracy (%) under different participation rate with Dirichlet-0.25 on CIFAR-100 dataset and small CNN network, where NTC indicates the number of total clients, and PR indicates the participation rate. Bold numbers indicate the best performance.

| NTC / PR (%) | 50 / 0.2 | 100 / 0.1 | 150 / 0.066 | 200 / 0.05 |
|---|---|---|---|---|
| FedSMU | 51.65 | **51.79** | **50.78** | **49.68** |
| FedAvg | 46.62 | 42.29 | 40.93 | 39.44 |
| FedEF-TopK | 47.25 | 42.69 | 40.23 | 37.41 |
| FedEF-Sign | 42.68 | 36.12 | 33.94 | 31.04 |
| FedLion | 48.41 | 48.33 | 47.81 | 48.74 |
| SCAFFOLD | **52.52** | 50.92 | 39.39 | 37.31 |
| SCALLION | 48.07 | 45.24 | 36.54 | 35.49 |

### 5.2.3 GENERALIZATION VS. DATA HETEROGENEITY

We further study the influence of data heterogeneity on the generalization performance of our FedSMU vs. FedAvg and SCAFFOLD. From resutls shown in Table 5, it is evident that the generalization of FedSMU surpasses FedAvg. By computing the top accuracy difference between the iid and Dirichlet-0.25 settings in Table 5, we observe a degradation of 4.13%, 6.61% and 4.29% in the top accuracy for FedSMU, FedAvg and SCAFFOLD, respectively. Consequently, FedSMU is affected less significantly by the degree of data heterogeneity.

Table 5: Top validation accuracy (%) under different data heterogeneity with 100 clients and 10% participation rate on CIFAR-100 dataset and small CNN network, where Dirichlet-0.25 indicates the highest heterogeneity and iid indicates the lowest heterogeneity.

| Algorithm | Dirichlet-0.25 | Dirichlet-0.6 | Dirichlet-0.8 | iid |
|---|---|---|---|---|
| FedSMU | 51.79 | 53.49 | 54.21 | 55.92 |
| FedAvg | 41.44 | 43.44 | 44.21 | 48.05 |
| SCAFFOLD | 50.49 | 50.02 | 53.89 | 54.78 |

Table 6: Top accuracy (%) comparison between ablation experiments with 100 clients and 10% participation, Dirichlet-0.25 on CIFAR-100 dataset and small CNN network, where NTC indicates the number of total clients, and PR indicates the participation rate.

| NTC / PR(%) | FedSMU | FedSMUMC | FedSMUM |
|---|---|---|---|
| 100 / 0.1 | 51.79 | 52.0 | 51.65 |
| 100 / 0.03 | 49.8 | 51.2 | 51.67 |

Besides, when horizontally comparing FedSMU and FedAvg, the improvement of FedSMU over FedAvg is 10.35%, 10.05%, 10.00%, and 7.87% for Dirichlet-0.25, Dirichlet-0.6, Dirichlet-0.8, and iid distributions, respectively. This indicates that FedSMU achieves a higher performance gain with the increasing degree of data heterogeneity. These results also validate that in highly heterogeneous data scenarios, where the difference

between clients' model updates becomes greater, our FedSMU can alleviate the local model heterogeneity through symbolic updates. This promotes the aggregation stability and improves generalization performance of the global model.

### 5.2.4 MEASURE OF GENERALIZATION

In the above experimental results, generalization refers to an algorithm's ability to achieve top test accuracy, where the test dataset is different from the training dataset. Furthermore, we consider an additional perspective on generalization to further evaluate the performance of our FedSMU algorithm. Here, generalization refers to a model's ability to achieve test performance at similar training error levels. Based on this definition, we compare the validation performance at similar training accuracy levels. The results in Table 7 show that on the CIFAR-10 and CIFAR-100 datasets with small CNN network, the FedSMU algorithm achieves the highest test accuracy and demonstrates the best generalization performance.

Table 7: Generalization performance comparison under various datasets with the small CNN network. Each table entry gives the average test accuracy on different training accuracy levels. "/" means it cannot reach the training accuracy. Bold numbers indicate the best performance.

| | Top-1 Test Accuracy (%). | | | | | | | |
|---|---|---|---|---|---|---|---|---|
| Dataset | Training Accuracy | FedAvg | SCAFFOLD | SCALLION | FedEF-HS | FedEF-TopK | FedEF-Sign | FedLion | FedSMU |
| CIFAR-10 | 83-84 | 75.14 | **77.49** | 75.9 | 75.63 | 75.48 | 75.7 | 77.22 | 77.1 |
| | 85-86 | 76.17 | 76.37 | 76.87 | 76.83 | 76.83 | 77.4 | 78.3 | **78.39** |
| | 87-88 | 78.56 | 79.24 | 77.79 | 77.85 | 77.74 | / | 79.23 | **79.78** |
| CIFAR-100 | 66-67 | 40.08 | 45.76 | 40.56 | / | 41.28 | / | 45.55 | **49.03** |
| | 68-69 | 40.6 | 46.16 | 40.88 | / | 41.81 | / | 46.15 | **50.04** |
| | 70-71 | 40.9 | 46.76 | 41.23 | / | 42.34 | / | 46.56 | **50.85** |

### 5.2.5 ABLATION STUDIES

Indeed, while our convergence analysis and experimental results demonstrate that FedSMU's performance is less affected by the client participation rate, the momentum of clients may still be extremely stale due to the partial participation in FL. In light of this, we design the two other variants, named FedSMUMC and FedSMUM, to examine the impact of this momentum staleness on the generalization performance. For FedSMUMC, clients upload 1-bit model updates along with extra momentum in the full precision. The server then aggregates that momentum to update the global momentum and broadcasts it at the next round as the initial momentum for the participating clients. For FedSMUM, every client, including the inactive clients, receives the updated global model. However, only the participating clients execute local training and upload model updates, while the other clients utilize the received global model to compute the momentum for the current round. See Appendix H for the detail of these two algorithms. Results in Table 6 indicate that by appropriately completing the momentum, we can marginally enhance the model performance, but it necessitates additional transmission of momentum with the full precision. Consequently in this sense, the local momentum staleness has a minimum impact on the global model's performance.

### 5.2.6 LIMITATION

Though our FedSMU effectively enhances the generalization performance while reducing the communication overhead, it may still have some limitations. First, our compression relies on the fixed symbol quantization, which might not be optimal for the adaptive scenarios. Exploring adaptive bit quantization further in our future research is promising to address this limitation. Second, our FedSMU necessitates additional caching of two sets of momentum, potentially resulting in a slightly increment of storage overhead. Last, from a theoretical perspective, though the generalization properties (Venkateswaran et al., 2023) of FedAvg under various assumptions have been extensively examined, such guarantees for the compression-based FL approaches remain an open problem.

## 6 CONCLUSION

In this paper, we have proposed the FedSMU algorithm that could effectively alleviate both the communication cost and data heterogeneity issues of federated learning. The key design was the symbolization of local client updates which were introduced to balance the contribution of each client and avoid the dominance by some relatively large update values. We carried out theoretical convergence analysis, and empirically showed that FedSMU converged faster to a higher top accuracy under the same communication cost. Under the condition of a very small partial client participation rate and relatively high data heterogeneity, FedSMU still demonstrated a better performance.

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

## A  DETAILED EXPERIMENT SETUP

We utilize the visual datasets including CIFAR-10 and CIFAR-100. CIFAR-10 and CIFAR-100 are two classic image classification datasets created by the Canadian Institute for Advanced Research (CIFAR). The CIFAR-10 dataset consists of 10 classes of images, with each class containing 6000 32x32-pixel color images. The CIFAR-100 dataset is an extension of CIFAR-10, containing 100 classes of images. These 100 classes are divided into 20 superclasses, each containing 5 subclasses. Each subclass contains 600 32x32-pixel color images. Both them comprise 50,000 images for training and 10,000 images for testing.

For CIFAR-10 and CIFAR-100, we employ a CNN model comprising two convolutional layers with sixty four $5 \times 5$ filters, two $2 \times 2$ max pooling layers, two fully connected layers with 384 and 192 neurons, and a softmax layer. We also used a larger network, ResNet18, to confirm that our algorithm still performs well in a larger network. ResNet18 contains 16 convolutional layers. These convolutional layers are distributed across several residual blocks, each containing two $3 \times 3$ convolutional layers. Additionally, there is a $7 \times 7$ convolutional layer at the beginning of the network. At the end of the network, there is a fully connected layer for output.

For Shakespeare dataset, we employ a RNN model. RNN consists of Input Layer (Receives the input at the current timestep), Hidden Layer (Receives the hidden state from the previous timestep along with the input at the current timestep to compute the new hidden state) and Output Layer (Outputs a result based on the current state of the hidden layer).

All approaches are implemented in PyTorch 1.4.0 and CUDA 9.2, with GEFORCE GTX 1080 Ti throughout our experiments.

In most federated learning scenarios, the total number of clients is set to 100 with the participation rate of 0.1, which is a classical experimental setting, like what FedLion (Tang & Chang, 2024) and FedAvg(McMahan et al., 2017) do. Therefore, with the simple CNN network, we set 100 clients with participation rates of 0.1 and 0.03 to verify the performance of our algorithm.

However, our computing resources (i.e., GEFORCE GTX 1080 Ti) are insufficient to support us to set 100 clients on the ResNet18 network, we thus can only select 10 clients and set the participation rate to 0.3.

We tune the hyper-parameter over a grid to compare the performance of different methods. For local update in all methods, we tune the local learning rate over $\{1, 0.1, 0.01, 0.001\}$ and set up 5 epochs of local updates with the minibatch $B = 50$.

For our proposed method FedSMU, we tune the parameter $\beta_1$ and $\beta_2$ over $\{0.9, 0.99, 0.999\}$, respectively, and set them both to 0.9 for CIFAR-10 and CIFAR-100 and 0.95 for Shakespeare. We tune the parameter $\gamma_1$ and $\gamma_2$ over $\{1, 0.1, 0.02, 0.018, 0.015, 0.013, 0.01, 0.005, 0.001\}$, respectively, since they are so sensitive, and set them to 0.015, 0.01 for CIFAR-10, 0.018, 0.01 for CIFAR-100 and 0.03, 0.1 for Shakespeare.

For FedLion (Tang & Chang, 2024), we tune the parameter $\beta_1$ and $\beta_2$ over $\{0.9, 0.99, 0.999\}$, respectively, and set $\beta_1$ to 0.9 and $\beta_2$ to 0.99 for CIFAR-10, CIFAR-100 and Shakespeare. We tune the parameter $\gamma_1$ over $\{1, 0.1, 0.01, 0.001, 0.0001\}$ and set it to 0.001 for CIFAR-10, CIFAR-100 and Shakespeare.

For FedEF-SGD (Li & Li, 2023), we tune the parameter $\eta_g$ over $\{1, 0.1, 0.01\}$ and set it to 1 for CIFAR-10,CIFAR-100 and Shakespeare.

For LocalLion and GlobalLion, we tune the parameter $\beta_1$ and $\beta_2$ over $\{0.9, 0.99, 0.999\}$, respectively, and set them to 0.9 and 0.99 for CIFAR-100. We tune the parameter $\gamma_1$ and $\gamma_2$ over $\{1, 0.1, 0.01, 0.001\}$, respectively, and set them to 0.001, 0.01 for CIFAR-100. We tune the parameter $\eta_g$ in LocalLion over $\{1, 0.1, 0.01, 0.001\}$ and set them to 1 for CIFAR-100.

For FedSMUM, we tune the parameter $\beta_1$ and $\beta_2$ over $\{0.9, 0.99, 0.999\}$, respectively, and set them to 0.9 and 0.99 for CIFAR-100. We tune the parameter $\gamma_1$ and $\gamma_2$ over $\{1, 0.1, 0.01, 0.001\}$, respectively, and set them to 0.01, 0.001 for CIFAR-100.

For FedSMUMC, we tune the parameter $\beta_1$ and $\beta_2$ over $\{0.9, 0.99, 0.999\}$, respectively, and set them both to 0.9 for CIFAR-100. We tune the parameter $\gamma_1$ and $\gamma_2$ over $\{1, 0.1, 0.01, 0.001\}$, respectively, and set them both to 0.01 for CIFAR-100.

## B  FURTHER EXPLANATION OF ALGORITHM 1

### B.1  DISCUSSION ON MOMENTUM

Our FedSMU algorithm requires each client to maintain and track its momentum state. In some extreme cases, where each client participates in the training only once during the entire training process, our algorithm may fail.

We do analyze this limitation from two perspectives. First, we test our algorithm at a lower client participation rate (i.e., where the momentum state of the clients is more stale). The results in Table 4 show that FedSMU still achieves a good accuracy in most cases. Additionally, we introduce a variant algorithm, FedSMUMC, which additionally store and communicate the complete momentum at the server at the cost of communication overhead. The results in Table 6 demonstrate that appropriately completing the momentum can marginally improve the model performance. Overall, the experimental results indicate that, in most cases, our FedSMU algorithm remains effective and does not fail due to the stale momentum.

Additionally, FedSMU only requires uploading the 1-bit model update after symbolization for each client and does not need to store the momentum terms on the server side. In contrast, FedLion (Tang & Chang, 2024) requires to aggregate and store momentum on the server side. Consequently, FedLion (Tang & Chang, 2024) necessitates the additional transmission of the full-precision momentum terms, along with the model update after quantization.

## B.2 DISCUSSION ON SIGN OPERATION

The "sign operation" refers to 1 bit per dimension, not a total of 1 bit for the entire model. In practice, the weights or gradients of each dimension in a model are typically stored as 32-bit floating-point values (float32). This means that each dimension requires 32 bits to upload. However, when applying the sign operation, each dimension only requires 1 bit to represent the sign, thus significantly reducing the communication overhead compared to using 32 bits per dimension.

In terms of the physical data transfer, the actual number of bits transmitted to the server can be calculated as: Total bits transferred $=$ (bits per dimension $\times$ the number of dimensions $\times$ the number of participating clients) $+$ the header size. For example, using a small CNN model with $n = 10$ clients, FedSMU would transfer approximately 0.95MB.

Though the service information for communication includes plenty of headers, for instance: the Ethernet header has 14 bytes for the header; IPv4 has 20 bytes header; TCP has header 20 header. Given the large volume of transmitted data, the header size of 54 bytes is negligible in comparison.

## B.3 DISCUSSION ON THE $\alpha$-BIT

First, we would like to acknowledge that for the general quantization-based compression algorithms, a higher precision quantization may often lead to a faster convergence. However, this may not hold for our FedSMU. This conclusion is based on the Lion optimizer (Chen et al., 2024) and our analysis, as follows.

1) The original Lion manuscript demonstrated that 1-bit quantization (via the sign operation) in centralized learning enhances the algorithm's convergence and generalization compared to other optimization techniques. That is, the authors analyzed that the sign operation introduces noise into the updates, which serves as a form of regularization, thereby improving both convergence and generalization.

2) Both experimentally (Figure 1) and intuitively, the symbolic operation (i.e., 1-bit quantization) helps alleviate the heterogeneity of model updates, as all updates have uniform magnitude across all dimensions for each client. Furthermore, reducing model heterogeneity should also intuitively contribute to improving model performance in heterogeneous federated settings.

3) However, such sign operation (e.g., QSGD) alone does not directly improve generalization in experiments. Inspired by Lion optimizer (Chen et al., 2024) that incorporates the sign operation and then enhances the convergence and generalization to learn in central learning, we introduce Lion's structure into federated learning and verify that this combination can indeed improve model generalization.

Consequently, we conclude that in our optimized structure, 1-bit quantization outperforms higher-bit or even full-bit quantization, since multi-bit compression does not guarantee that the update amplitude of each client is consistent. Experimental results in Table 8 further validate that for our designed optimization algorithm, using a higher-bit compression does not enhance the algorithm's convergence or generalization.

## B.4 DISCUSSION ON SERVER-TO-CLIENT COMPRESSION

Table 8: Number of communication rounds to achieve a preset target test accuracy with Dirichlet-0.25 on CIFAR-10 dataset and small CNN network. "/" means it caanot reach the test accuracy and bold numbers indicate the smallest rounds.

| Number of rounds needed for achieving a target test accuracy. | | | |
| --- | --- | --- | --- |
| Test Accuracy(%) | 1-bit (FedSMU) | 3-bit | 8-bit |
| 40 | **26** | 65 | 54 |
| 45 | **33** | 91 | 68 |
| 50 | **50** | 168 | 119 |
| 55 | **65** | 285 | 185 |
| 60 | **118** | 456 | 224 |
| 65 | **288** | 833 | 342 |
| 67.5 | **399** | 1260 | 401 |
| 69 | 507 | 1967 | **456** |
| 72.3 | **642** | / | 643 |
| 75 | 1142 | / | **1040** |
| 77.5 | **1746** | / | 1979 |

Due to the partial participation characteristic of federated learning, the server must broadcast the new global model, rather than a simple global model update, to initialize newly participating clients. This limitation prevents the direct application of uploaded model update compression techniques in our FedSMU to the downloaded global model.

We will consider some model lightweight techniques, such as mixed-precision model compression, as a promising future research direction to compress the server-to-client communication in our FedSMU algorithm.

## C   FURTHER EXPLANATION OF ASSUMPTION 4.1 TO ASSUMPTION 4.3

Assumption 4.1 is a standard assumption in the convergence analysis of optimization algorithms. Similar assumptions can be found in MARINA (Gorbunov et al., 2021) ("Assumption 1.1") and EF21 (Richtárik et al., 2021) ("Assumption 1").

For Assumption 4.2, it is fundamental for SGD-based optimization algorithms. In SGD, model updates are computed using mini-batch sampling rather than full-batch, under the assumption that the sampling process is unbiased and accounts for the variance introduced. However, MARINA(Gorbunov et al., 2021) bypasses this assumption by employing the Gradient Descent (GD) instead of Stochastic Gradient Descent (SGD), eliminating the variance caused by mini-batch sampling. EF21(Richtárik et al., 2021) discuss this stochastic gradient explicitly in Section "F: Dealing with Stochastic Gradients (Details for Section 3.6)".

For Assumption 4.3, we acknowledge that this is a stronger assumption, ensuring that both the compressed targets and the momentum term (the moving averages of gradients) in our theoretical analysis are bounded. Moreover, this assumption has been adopted in other federated optimization and sign-based compression studies (Sun et al., 2023; Reddi et al., 2020). Specifically, in (Sun et al., 2023), the authors demonstrate the convergence of distributed SIGNSGD with momentum under Assumption 4, which is also the bounded gradient assumption.

Regarding $y = \frac{x^2}{2}$ over $\mathbb{R}$, it is true that the gradient does not have an upper bound for the domain $\mathbb{R}$. However, in neural networks, the input domain is typically bounded, ensuring the gradient is also bounded. For instance, gradient clipping is a commonly used technique to control the gradient's upper bound and prevent gradient explosion.

# D  A PROOF OF THEOREM 4.4

*Proof.* Set $\Delta_t = \frac{1}{n}\sum_{i=1}^{n} u_t^i = \frac{1}{n}\sum_{i=1}^{n} sign[\beta_1 m_{t-1}^i + (1-\beta_1)g_t^i]$ and $||\Delta_t||^2 = \sum_{j=1}^{d} |\Delta_t^j|^2$, where $d$ is the dimensions of parameters.

Since $\gamma_2$ is adjustable, so for each coordinate $j$, we can assume $|\gamma_2 x_t^j| \le 1$. ($\|\gamma_2 x\|_\infty \le 1$ )

It has been demonstrated in (Chen et al., 2023) (Equation 5 on Page 3). (Chen et al., 2023) clarifies in Abstract that "Lion is a theoretically novel and principled approach for minimizing a general loss function f(x) while enforcing a bound constraint $||x||_\infty \le \frac{1}{\gamma_2}$." Here $\gamma_2$ is the weight decay coefficient and $x_t$ is the model. Such an assumption has also been used in another algorithm (Liu et al., 2024) based on Lion optimizer (Equation 7 on Page 4).

And thus $|\Delta_t^j - \gamma_2 x_t^j| \le |\Delta_t^j| + |\gamma_2 x_t^j| \le 2$. Then $||\Delta_t||^2 \le d$ and $||\Delta_t - \gamma_2 x_t||^2 \le 4d$.

With Assumption 4.1, we have

$$
\begin{aligned}
f(x_{t+1}) &\le f(x_t) + \langle \nabla f(x_t), x_{t+1} - x_t \rangle + \frac{L}{2}||x_{t+1} - x_t||^2 \\
&= f(x_t) + \langle \nabla f(x_t), \gamma_1 \Delta_t - \gamma_1 \gamma_2 x_t \rangle + \frac{L}{2}||\gamma_1 \Delta_t - \gamma_1 \gamma_2 x_t||^2 \\
&= f(x_t) - \langle \nabla f(x_t), \gamma_1 sign(\nabla f(x_t)) \rangle + \langle \nabla f(x_t), \gamma_1 \Delta_t - \gamma_1 \gamma_2 x_t + \gamma_1 sign(\nabla f(x_t)) \rangle \quad (4) \\
&\quad + \frac{L}{2}||\gamma_1 \Delta_t - \gamma_1 \gamma_2 x_t||^2 \\
&= f(x_t) - \gamma_1 ||\nabla f(x_t)||_1 + \gamma_1 \underbrace{\langle \nabla f(x_t), \Delta^t - \gamma_2 x_t + sign(\nabla f(x_t)) \rangle}_{A} + 2L\gamma_1^2 d.
\end{aligned}
$$

Considering the calculation of $A$:

$$
\begin{aligned}
A &= \langle \nabla f(x_t), \Delta_t - \gamma_2 x_t + sign(\nabla f(x_t)) \rangle \\
&= \langle \nabla f(x_t), \frac{1}{n}\sum_{i=1}^{n} u_t^i - \gamma_2 x_t + sign(\nabla f(x_t)) \rangle,
\end{aligned} \quad (5)
$$

For any dimension $j$, assume $|\gamma_2 x_t^j| < 1$, and then we have $\nabla f(x_t^j)(\frac{1}{n}\sum_{i=1}^{n} u_t^{i,j} - \gamma_2 x_t^j + sign(\nabla f(x_t^j))) \le 3|\nabla f(x_t^j)| = 3G|\frac{1}{G}\nabla f(x_t^j)| < 3G|\frac{1}{G}\nabla f(x_t^j) + \gamma_1 sign(\nabla f(x_t^j))|$.

So,

$$
\begin{aligned}
A &< 3G||\frac{1}{G}\nabla f(x_t) + \gamma_1 sign(\nabla f(x_t))||_1 \\
&\le 3\sqrt{d}G||\frac{1}{G}\nabla f(x_t) + sign(\nabla f(x_t))||.
\end{aligned} \quad (6)
$$

Substitute Eq. (6) into Eq. (4), we further have

$$
\begin{aligned}
f(x_{t+1}) - f(x_t) &\le -\gamma_1 ||\nabla f(x_t)||_1 + \gamma_1 \underbrace{\langle \nabla f(x_t), \Delta_t - \gamma_2 x_t + sign(\nabla f(x_t)) \rangle}_{A} + 2L\gamma_1^2 d \\
&\le -\gamma_1 ||\nabla f(x_t)||_1 + \gamma_1 3\sqrt{d}G \underbrace{||\frac{1}{G}\nabla f(x_t) + \gamma_1^2 sign(\nabla f(x_t))||}_{B} + 2L\gamma_1^2 d.
\end{aligned} \quad (7)
$$

Taking the expectation of $B$, with Assumption 4.3 we have

$$
\begin{aligned}
E(B) &\le E(||\underbrace{\frac{1}{G}\nabla f(x_t) + \frac{\gamma_1}{nKG}\sum_{i=1}^{n} v_t^i}_{\epsilon_t}||) + E(||\gamma_1 sign(\nabla f(x_t)) - \frac{\gamma_1}{nKG}\sum_{i=1}^{n} v_t^i||) \\
&\le E(||\epsilon_t||) + E(\gamma_1 \sqrt{\sum_{j=1}^{d} |sign(\nabla f(x_t^j)) - \frac{1}{n}\sum_{i=1}^{n} \frac{1}{KG} v_t^{i,j}|^2}) \\
&\le E(||\epsilon_t||) + 2\gamma_1 \sqrt{d}.
\end{aligned} \quad (8)
$$

Taking the expectation of Eq. (8), we have

$$
E(f(x_{t+1})) - E(f(x_t)) \le -\gamma_1 E(||\nabla f(x_t)||_1) + 3\sqrt{d}\gamma_1 GE(||\epsilon_t||) + 6d\gamma_1^2 G + 2L\gamma_1^2 d. \quad (9)
$$

$$\epsilon_t = \frac{1}{G}\nabla f(x_t) + \frac{\gamma_1}{nKG}\sum_{i=1}^{n} v_t^i$$

$$= \frac{1}{Gn}\sum_{i=1}^{n}\nabla F_i(x_t) + \frac{\gamma_1}{nKG}\sum_{i=1}^{n} v_t^i$$

$$= \frac{1}{n}\sum_{i=1}^{n}(\frac{1}{G}\nabla F_i(x_t) + \frac{\gamma_1}{KG}v_t^i) \tag{10}$$

$$= \frac{1}{n}\sum_{i=1}^{n}\epsilon_t^i$$

Further define $h_t^i = -\frac{1}{KG}\sum_{k=1}^{K}\nabla F_i(y_{t,k}^i;\xi_{t,k}^i)$, $\delta_t^i = h_t^i + \frac{1}{G}\nabla F_i(x_t)$.

Referring to Algorithm 1, we have $v_t^i = \beta_2 v_{t-\tau^i}^i + (\beta_1 - \beta_2)g_{t-\tau^i}^i + (1-\beta_1)g_t^i$.

For each client $i$, we have

$$\frac{\gamma_1}{KG}v_t^i = \frac{\gamma_1}{KG}\beta_2 v_{t-\tau^i}^i + \gamma_1\eta(\beta_1-\beta_2)h_{t-\tau^i}^i + \gamma_1\eta(1-\beta_1)h_t^i$$

$$= \beta_2(\epsilon_{t-\tau^i}^i - \frac{1}{G}\nabla F_i(x_{t-\tau^i})) + \gamma_1\eta(\beta_1-\beta_2)(\delta_{t-\tau^i}^i - \frac{1}{G}\nabla F_i(x_{t-\tau^i})) \tag{11}$$

$$+ \gamma_1\eta(1-\beta_1)(\delta_t^i - \frac{1}{G}\nabla F_i(x_t)).$$

Converting the form of Eq. (11), we have

$$\epsilon_t^i = \beta_2\epsilon_{t-\tau^i}^i + \gamma_1\eta(\beta_1-\beta_2)\delta_{t-\tau^i}^i + \gamma_1\eta(1-\beta_1)\delta_t^i + \underbrace{\frac{1}{G}[\nabla F_i(x_t) - \nabla F_i(x_{t-\tau^i})]}$$

$$+ \underbrace{[\frac{-\gamma_1\eta + \gamma_1\eta\beta_1}{G}\nabla F_i(x_t) - \frac{\beta_2 - 1 + \gamma_1\eta(\beta_1-\beta_2)}{G}\nabla F_i(x_{t-\tau^i})]}_{s_t^i}. \tag{12}$$

Taking the $\ell_2$ norm of $s_t^i$ and assuming $\beta_1 \le \beta_2$, with Assumption 4.1 and Assumption 4.3, we have

$$||s_t^i|| = \frac{1}{G}||\nabla F_i(x_t) - \nabla F_i(x_{t-\tau^i})||$$

$$+ ||\frac{-\gamma_1\eta + \gamma_1\eta\beta_1}{G}\nabla F_i(x_t)|| + ||\frac{\beta_2 - 1 + \gamma_1\eta(\beta_1-\beta_2)}{G}\nabla F_i(x_{t-\tau^i})||$$

$$= \frac{1}{G}||\nabla F_i(x_t) - \nabla F_i(x_{t-\tau^i})||$$

$$+ \frac{\gamma_1\eta(1-\beta_1)}{G}||\nabla F_i(x_t)|| + \frac{1-\beta_2 + \gamma_1\eta(\beta_2-\beta_1)}{G}||\nabla F_i(x_{t-\tau^i})|| \tag{13}$$

$$\le \frac{1}{G}||\nabla F_i(x_t) - \nabla F_i(x_{t-\tau^i})||$$

$$+ \frac{\gamma_1\eta(1-\beta_1)}{G}||\nabla F_i(x_t)|| + \frac{1-\beta_1 + \gamma_1\eta(\beta_2-\beta_1)}{G}||\nabla F_i(x_{t-\tau^i})||$$

$$\le \frac{L}{G}||x_t - x_{t-\tau^i}|| + \gamma_1\eta(1-\beta_1) + 1 - \beta_1 + \gamma_1\eta(\beta_2-\beta_1)$$

$$\le \frac{2\gamma_1 L\sqrt{d}\tau^i}{G} + \gamma_1\eta(1+\beta_2) + 1 - \beta_1.$$

Taking the expectation of $||\delta_t^i||^2$ and using the Lemma D.1, we have

$$E(||\delta_t^i||^2) = E(||-\frac{1}{KG}\sum_{k=1}^{K}\nabla F_i(y_{t,k}^i;\xi_{t,k}^i) + \frac{1}{G}\nabla F_i(x_t)||^2)$$

$$\le \frac{\sum_{k=1}^{K}E||\nabla F_i(y_{t,k}^i;\xi_{t,k}^i) - \nabla F_i(x_t)||^2}{K^2G^2}$$

$$\le \frac{L^2\sum_{k=1}^{K}E||y_{t,k}^i - x_t||^2}{K^2G^2} \tag{14}$$

$$\le \frac{L^2(8K\eta^2\sigma_l^2 + 16K^2\eta^2\sigma_l^2 + 16K^2\eta^2G^2)}{KG^2}.$$

Taking the $\ell_2$ norm of Eq. (12) and using Eq. (13), let $\tau_0^i = 0, \tau_1^i = \tau^i, \sum_{j=0}^c \tau_j^i = \tau_{c^i}, \max\{\tau_j^i\}_{1 \le j \le c+1} = \tau_{max}^i, \tau_{c^i} > t - 1, c = c^i \le t - 1$ and we have

$$
\begin{aligned}
||\epsilon_t^i|| &\le ||\beta_2 \epsilon_{t-\tau^i}^i|| + ||s_t^i|| + ||\gamma_1(\beta_1 - \beta_2)\delta_{t-\tau^i}^i|| + ||\gamma_1(1 - \beta_1)\delta_t^i|| \\
&= ||\beta_2^{c^i+1}\epsilon_0^i|| + ||\sum_{j=0}^{c^i} \beta_2^j s_{t-\tau_j}^i|| + ||\gamma_1(\beta_1 - \beta_2)\sum_{j=0}^{c^i} \beta_2^j \delta_{t-\tau_{j+1}}^i|| \\
&\quad + ||\gamma_1(1 - \beta_1)\sum_{j=0}^{c^i} \beta_2^j \delta_{t-\tau_j}^i|| \\
&\le \beta_2^{c^i+1}||\epsilon_0^i|| + \left(\frac{2\gamma_1 L\sqrt{d}\tau_{max}^i}{G} + \gamma_1\eta(1 + \beta_2) + 1 - \beta_1\right)\sum_{j=0}^{c^i} \beta_2^j \\
&\quad + \gamma_1(\beta_1 - \beta_2)||\sum_{j=0}^{c^i} \beta_2^j \delta_{t-\tau_{j+1}}^i|| + \gamma_1(1 - \beta_1)||\sum_{j=0}^{c^i} \beta_2^j \delta_{t-\tau_j}^i|| \\
&\le \beta_2^{c^i+1}||\epsilon_0^i|| + \frac{2\gamma_1 L\sqrt{d}\tau_{max}^i}{G(1 - \beta_2)} + \frac{\gamma_1\eta(1 + \beta_2) + 1 - \beta_1}{1 - \beta_2} + \gamma_1(\beta_1 - \beta_2)||\sum_{j=0}^{c^i} \beta_2^j \delta_{t-\tau_{j+1}}^i|| \\
&\quad + \gamma_1(1 - \beta_1)||\sum_{j=0}^{c^i} \beta_2^j \delta_{t-\tau_j}^i||.
\end{aligned}
\tag{15}
$$

Notice that the random variables $\left(\delta_t^i\right)_{1 \le t \le T}$ are independent, so $\mathbb{E}\left\langle \delta_{t1}^i, \delta_{t2}^i \right\rangle = 0$. Take the expectation of $||\sum_{j=0}^{c^i} \beta_2^j \delta_{t-\tau_j}^i||, ||\sum_{j=0}^{c^i} \beta_2^j \delta_{t-\tau_{j+1}}^i||$ and use the Eq. (14), we have

$$
\begin{aligned}
E||\sum_{j=0}^{c^i} \beta_2^j \delta_{t-\tau_j}^i|| = E||\sum_{j=0}^{c^i} \beta_2^j \delta_{t-\tau_{j+1}}^i|| &\le \sqrt{E(||\sum_{j=0}^{c^i} \beta_2^j \delta_{t-\tau_j}^i||^2)} \\
&= \sqrt{E(\sum_{j=0}^{c^i} \beta_2^{2j}||\delta_{t-\tau_j}^i||^2)} \\
&\le \sqrt{\frac{1}{1 - \beta_2^2}\frac{L^2(8K\eta^2\sigma_l^2 + 16K^2\eta^2\sigma_l^2 + 16K^2\eta^2 G^2)}{KG^2}}.
\end{aligned}
\tag{16}
$$

Taking the expectation of Eq. (15) and substituting Eq. (16) in it, we further have

$$
\begin{aligned}
E||\epsilon_t^i|| &\le \beta_2^{c^i+1}||\epsilon_0^i|| + \frac{2\gamma_1 L\sqrt{d}\tau_{max}^i}{G(1 - \beta_2)} + \frac{\gamma_1\eta(1 + \beta_2) + 1 - \beta_1}{1 - \beta_2} \\
&\quad + \gamma_1(1 - \beta_2)\sqrt{\frac{1}{1 - \beta_2^2}\frac{L^2(8K\eta^2\sigma_l^2 + 16K^2\eta^2\sigma_l^2 + 16K^2\eta^2 G^2)}{KG^2}}.
\end{aligned}
\tag{17}
$$

Recursively iterating it from $t = 0$ to $t = T$ and substituting Eq. (17) into Eq. (10), we have

$$\frac{1}{T}\sum_{t=1}^{T} E(||\nabla f(x_t)||_1) \leq \frac{f(x_0) - minf}{\gamma_1 T} + 3\sqrt{d}GE(||\epsilon_t||) + 6d\gamma_1 G + 2L\gamma_1 d$$

$$\leq \frac{f(x_0) - minf}{\gamma_1 T} + 3\sqrt{d}G\frac{\sum_{t=1}^{T}\sum_{i=1}^{n} E||\epsilon_t^i||}{nT} + 6d\gamma_1 G + 2L\gamma_1 d$$

$$\leq \frac{f(x_0) - minf}{\gamma_1 T} + 3\sqrt{d}G\frac{\sum_{t=1}^{T}\sum_{i=1}^{n} \beta_2^{c^i+1}||\epsilon_0^i||}{nT} + \frac{3\sqrt{d}G 2\gamma_1 L\sqrt{d}\tau_{max}}{G(1-\beta_2)}$$

$$+ \frac{3\sqrt{d}G\gamma_1\eta(1+\beta_2) + 3\sqrt{d}G(1-\beta_1)}{1-\beta_2}$$

$$+ 3\sqrt{d}G\gamma_1(1-\beta_2)\sqrt{\frac{1}{1-\beta_2^2}\frac{L^2(8K\eta^2\sigma_l^2 + 16K^2\eta^2\sigma_l^2 + 16K^2\eta^2 G^2)}{KG^2}} \quad (18)$$

$$+ 6d\gamma_1 G + 2L\gamma_1 d$$

$$\leq \frac{f(x_0) - minf}{\gamma_1 T} + \frac{3\sqrt{d}G\phi}{nT(1-\beta_2)} + \frac{6dL\tau_{max}\gamma_1}{1-\beta_2}$$

$$+ \frac{3\sqrt{d}G\gamma_1\eta(1+\beta_2) + 3\sqrt{d}G(1-\beta_1)}{1-\beta_2}$$

$$+ 6\sqrt{d}L\eta\gamma_1\sqrt{\frac{1-\beta_2}{1+\beta_2}(2\sigma_l^2 + 4K\sigma_l^2 + 4KG^2)}$$

$$+ 6d\gamma_1 G + 2L\gamma_1 d.$$

where $\tau_{max} = \max\{\tau_{max}^i\}_{1\leq i\leq m}$, $\phi = \sum_{i=1}^{m} ||\epsilon_0^i||$ when $1 \leq t \leq T$.

Finally, when $\gamma_1 = \frac{1}{L\sqrt{T}}$ and $1 - \beta_1 = \frac{1}{\sqrt{T}}$, we complete the proof that

$$\frac{1}{T}\sum_{t=1}^{T} E(||\nabla f(x_t)||_1) \leq \frac{L(f(x_0) - minf)}{\sqrt{T}} + \frac{3\sqrt{d}G\phi}{nT(1-\beta_2)} + \frac{6d\tau_{max}}{(1-\beta_2)\sqrt{T}}$$

$$+ \frac{3\sqrt{d}G\eta(1+\beta_2)}{(1-\beta_2)L\sqrt{T}} + \frac{3\sqrt{d}G}{(1-\beta_2)\sqrt{T}} \quad (19)$$

$$+ \frac{6\sqrt{d}\eta}{\sqrt{T}}\sqrt{\frac{1-\beta_2}{1+\beta_2}(2\sigma_l^2 + 4K\sigma_l^2 + 4KG^2)}$$

$$+ \frac{6dG}{L\sqrt{T}} + \frac{2d}{\sqrt{T}}.$$

$\square$

**Lemma D.1.** *Let Assumption 4.1, Assumption 4.2 and Assumption 4.3 hold for $\xi_t^i$ and $\nabla F_i(\cdot; \cdot)$. Assume node $i$ performs local SGD as*

$$y_{t,k}^i = y_{t,k-1}^i - \eta\nabla F_i(y_{t,k-1}^i, \xi_{t,k-1}^i)$$

*with $y_{t,0}^i = x_t$. Like the lemma proved in (Sun et al., 2023), since $0 < \eta \leq \frac{1}{4LK}$, it holds*

$$\mathbb{E}\left\|y_{t,k}^i - x_t\right\|^2 \leq 8K\eta^2\sigma_l^2 + 16K^2\eta^2\sigma_l^2 + 16K^2\eta^2 G^2$$

*Proof.*

Following the proof in (Sun et al., 2023), note that for any $k \in \{1, \ldots, K\}$, in node $i$, (20)

$$\mathbb{E}\left\|y_{t,k}^i - x_t\right\|^2 = \mathbb{E}\left\|y_{t,k-1}^i - \eta\nabla F_i(y_{t,k-1}^i, \xi_{t,k-1}^i) - x_t\right\|^2$$

$$\leq \mathbb{E}\|y_{t,k-1}^i - x_t - \eta\big(\nabla F_i\left(y_{t,k-1}^i; \xi_{t,k-1}^i\right)$$

$$- \nabla F_i\left(y_{t,k-1}^i\right) + \nabla F_i\left(y_{t,k-1}^i\right)$$

$$- \nabla F_i\left(x_t\right) + \nabla F_i\left(x_t\right)\big)\|^2. \quad (21)$$

By using the Cauchy's inequality

$$\mathbb{E}\|\mathbf{a} + \mathbf{b}\|^2 \leq \left(1 + \frac{1}{\psi}\right)\mathbb{E}\|\mathbf{a}\|^2 + (1 + \psi)\mathbb{E}\|\mathbf{b}\|^2$$

with $a = y_{t,k-1}^i - x_t - \eta\left(\nabla F_i\left(y_{t,k-1}^i; \xi_{t,k-1}^i\right) - \nabla F_i\left(y_{t,k-1}^i\right)\right), b = \eta\left(\nabla F_i\left(y_{t,k-1}^i\right) - \nabla F_i\left(x_t\right) + \nabla F_i\left(x_t\right)\right)$ and $\psi = 2K - 1$.

Denote that $\Re := \left(1 + \frac{1}{2K-1}\right)\mathbb{E}\|y_{t,k-1}^i - x_t - \eta\left(\nabla F_i\left(y_{t,k-1}^i; \xi_{t,k-1}^i\right) - \nabla F_i\left(y_{t,k-1}^i\right)\right)\|^2$. and $\Im := 2K\eta^2\mathbb{E}\|\nabla F_i\left(y_{t,k-1}^i\right) - \nabla F_i\left(x_t\right) + \nabla F_i\left(x_t\right)\|^2$. The unbiased expectation property of $\nabla F_i\left(y_{t,k-1}^i; \xi_{t,k}^i\right)$ gives us

$$\Re = \left(1 + \frac{1}{2K-1}\right)\left(\mathbb{E}\left\|y_{t,k-1}^i - x_t\right\|^2 + \eta^2\mathbb{E}\left\|\nabla F_i\left(y_{t,k-1}^i; \xi_{t,k-1}^i\right) - \nabla F_i\left(y_{t,k-1}^i\right)\right\|^2\right)$$

$$\leq \left(1 + \frac{1}{2K-1}\right)\left(\mathbb{E}\left\|y_{t,k-1}^i - x_t\right\|^2 + \eta^2\sigma_l^2\right)$$

On the other hand, we have the following bound

$$\Im \leq 4K\eta^2\mathbb{E}\left\|\nabla F_i\left(y_{t,k-1}^i\right) - \nabla F_i\left(x_t\right)\right\|^2 + 4K\eta^2\mathbb{E}\left\|\nabla F_i\left(x_t\right)\right\|^2$$

$$\leq 4L^2K\eta^2\mathbb{E}\left\|y_{t,k-1}^i - x_t\right\|^2 + 4K\eta^2G^2$$

When $0 < \eta \leq \frac{1}{4LK}$,

$$1 + \frac{1}{2K-1} + 4L^2K\eta^2 \leq 1 + \frac{1}{K-1}$$

and we can obtain

$$\mathbb{E}\left\|y_{t,k}^i - x_t\right\|^2$$

$$\leq \left(1 + \frac{1}{2K-1} + 4L^2K\eta^2\right)\mathbb{E}\left\|y_{t,k-1}^i - x_t\right\|^2 + 2\eta^2\sigma_l^2 + 4K\eta^2\sigma_l^2 + 4K\eta^2G^2$$

$$\leq \left(1 + \frac{1}{K-1}\right)\mathbb{E}\left\|y_{t,k-1}^i - x_t\right\|^2 + 2\eta^2\sigma_l^2 + 4K\eta^2\sigma_l^2 + 4K\eta^2G^2$$

The recursion from $j = 0$ to $K$ yields

$$\mathbb{E}\left\|y_{t,k}^i - x_t\right\|^2 \leq \sum_{j=0}^{K-1}\left(1 + \frac{1}{K-1}\right)^j\left[2\eta^2\sigma_l^2 + 4K\eta^2\sigma_l^2 + 4K\eta^2G^2\right]$$

$$\leq (K-1)\left[\left(1 + \frac{1}{K-1}\right)^K - 1\right] \times \left[2\eta^2\sigma_l^2 + 4K\eta^2\sigma_l^2 + 4K\eta^2G^2\right]$$

$$\leq 8K\eta^2\sigma_l^2 + 16K^2\eta^2\sigma_l^2 + 16K^2\eta^2G^2$$

where we used the inequality $\left(1 + \frac{1}{K-1}\right)^K \leq 5$ holds for any $K \geq 1$. □

# E    FURTHER EXPLORATION OF THEOREM 4.4

## E.1    DISCUSSION ON THE $L_1$ FORM

The convergence rate in terms of $L_1$ norm with a dependency of $d$ is common in sign-based method, like Theorem 4 in (Sun et al., 2023), p.6, Theorem 1 in (Chen et al., 2020), P.6, and Equation 35 in (Jin et al., 2020b), P.19. Their form of expression is similar to Theory 4.4. However, it is worth noting that the value of the learning rate $\gamma_1$ may be critical. In (Chen et al., 2020) and (Jin et al., 2020b), the authors use $\gamma_1 = \frac{1}{\sqrt{Td}}$ to reduce the power of $d$. In that way, if we take $\gamma_1 = \frac{1}{L\sqrt{Td}}$, Theorem 4.4 can only depend on $\sqrt{d}$.

## E.2    COMPARISON WITH FEDLION (TANG & CHANG, 2024)

In this section, we highlight some key differences between our algorithm and FedLion. Specifically, while FedLion (Tang & Chang, 2024) does not rely on the Assumption 4.3, it also introduces an additional stronger assumption (A.4 in the original manuscript of FedLion(Tang & Chang, 2024)):" There exists a constant $\alpha \leq \frac{1}{3}$, such that $\|\nabla f(x) - \nabla f_i(x)\|_1 \leq \alpha \|\nabla f(x)\|_1, \forall i, x$."

Additionally, FedLion (Tang & Chang, 2024) cannot be reduced to a standard Lion optimizer, because it merely parallelizes the execution of the Lion optimizer on the client side and incorporates multi-precision quantization for communication compression. Therefore, even when $K = 1$ and $m = 1$, it does not degrade to a standard Lion optimizer, and its convergence rate cannot be directly applied to the Lion optimizer.

In contrast, when $K = 1$ and $m = 1$ in our algorithm, it degenerates to the standard Lion optimizer, and its convergence rate can be applied to the Lion optimizer.

## E.3    DETERMINANTS OF THE CONVERGENCE BOUND

In the theoretical analysis of FedSMU, $\tau_{max}$ and $K$ control the client participation rate and the number of local updates, respectively. The convergence rate in Theorem 4.4 demonstrates that increasing the client participation rate and reducing the number of local updates can tighten the convergence bound.

It is intuitive that a increased participation rate can improve the model's convergence rate. A higher client participation ratio helps the global model gather more information from the clients, which reduces the overfitting and mitigates the impact of local data heterogeneity, thereby improving the convergence bound. This conclusion is also supported by Table 1 in (McMahan et al., 2017).

On the other hand, in the convergence upper bound of Theorem 4.4, the term $4K\sigma_l^2 + 4KG^2$ indicates that more local updates amplify both the local sampling variance $\sigma_l$ and the gradients accumulation $G$, resulting in a slower convergence. This aligns with our intuition. For the SGD-based local updates, as the number of local updates increases, the cumulative sampling variance $\sigma_l$ and gradients $G$ grows, also further leading to more divergent update directions caused by data heterogeneity. This negative effect is also experimentally demonstrated in Page 7 of (McMahan et al., 2017), where Figure 3 strongly states that "for very large numbers of local epochs, FedAvg can plateau or diverge."

## E.4    FACTORS INFLUENCING CONVERGENCE SPEED

Theoretically, a lower client participation rate (i.e., a larger $\tau_{max}$) leads to a slower algorithm convergence. Similarly, a higher model dimension $d$ also results in a slower algorithm convergence. To validate this, we have conducted the following experiments.

All of these experiments are done on CIFAR-10 dataset with Dirichlet-0.25. To illustrate the relationship between the model dimension and convergence rate, we use two different CNN models ($d1 = 797248$ and $d2 = 1723648$) to study the impact of model dimension. Note that here we modify the size of the convolutional layers, keeping the model depth constant. The number of clients is 100 with the participation ratio of 0.1. To illustrate the relationship between participation rate and convergence rate, We use the CNN network with $d1 = 797248$ and set the number of clients as 100 with different participation ratio ($\frac{n}{m} = 0.03$ and $\frac{n}{m} = 0.1$, represented in the Table 9 by L and H) to demonstrate the influence of client participation rate.

The experimental results in Table 9 show that when the participation rate is higher (i.e., the $\tau_{max}$ is smaller) and the dimension is smaller, the convergence speed can be faster, which matches with the Theorem 4.4 and is intuitional.

Table 9: Number of communication rounds to achieve a preset target test accuracy with Dirichlet-0.25 on CIFAR-10 dataset and different CNN network. d1 and d2 indicate small and large dimension while L and H indicate low and high participation rates. Bold numbers indicate the smallest rounds.

| Number of rounds needed for achieving a target test accuracy. | | | |
|---|---|---|---|
| Test Accuracy(%) | d1, H | d2, H | d1, L |
| 40 | **26** | 30 | 46 |
| 45 | **33** | 35 | 112 |
| 50 | 50 | **43** | 193 |
| 55 | **65** | 65 | 226 |
| 60 | 118 | **111** | 384 |
| 65 | 288 | **275** | 899 |
| 67.5 | **399** | 409 | 949 |
| 69 | **507** | 507 | 1025 |
| 72.3 | **642** | 769 | 1752 |
| 75 | **1142** | 1185 | 2221 |
| 77.5 | 1746 | **1745** | 2878 |

# F    FURTHER EXPLORATION OF CONVERGENCE SPEED

## F.1    CONVERGENCE PERFORMANCE VS. COMMUNICATION ROUNDS

In Figure 3, we show the convergence performance in terms of communication rounds. Also, we compare the convergence rate using the number of communication rounds required to achieve the target accuracy and the results are presented in the Table 10.

For CIFAR-100 and Shakespeare, our algorithm does not require more communication rounds compared to most algorithms. However, for CIFAR-10, it slightly exceeds the number of rounds needed by other algorithms. This may be attributed to the simplicity of the CIFAR-10 dataset, which has a lower degree of heterogeneity. In this case, our algorithm's strengths are not fully utilized, as the initial training process is already close to convergence.

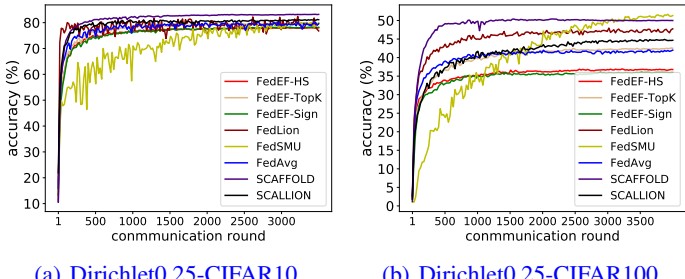

(a) Dirichlet0.25-CIFAR10          (b) Dirichlet0.25-CIFAR100

Figure 3: Convergence performance vs. number of communication rounds on CIFAR-10 and CIFAR-100, with 100 clients and 10% participation, using small CNN network for different algorithms.

## F.2    CONVERGENCE PERFORMANCE VS. WALL-CLOCK

We test the wall-clock time needed for each baseline to execute one communication round. Take CIFAR-100 and participation rate $\frac{n}{m} = 0.1$ as an example, the average wall-clock time required to execute a round is

Table 10: Number of communication rounds to achieve a preset target accuracy with 100 clients and 10% participation. CIFAR-10 and CIFAR-100 use the small CNN network and Shakespeare uses the RNN network. "/" means it cannot reach the training accuracy. Bold numbers indicate the best performance.

| Number of communication rounds to achieve a preset target accuracy. | | | | | | | | |
|---|---|---|---|---|---|---|---|---|
| Dataset | Training Accurac (%) | FedAvg | SCAFFOLD | SCALLION | FedEF-HS | FedEF-TopK | FedEF-Sign | FedLion | FedSMU |
| CIFAR-10 (Dir0.25) | 55 | 26 | 29 | 23 | 41 | 37 | 50 | **21** | 65 |
| | 60 | 43 | 43 | **29** | 62 | 57 | 81 | 33 | 118 |
| | 65 | 57 | 62 | 48 | 108 | 96 | 111 | **50** | 288 |
| CIFAR-100 (Dir0.25) | 35 | 193 | **86** | 286 | 690 | 355 | 794 | 100 | 832 |
| | 40 | 629 | **142** | 730 | / | 882 | / | 225 | 1218 |
| | 45 | / | **270** | 3703 | / | / | / | 632 | 1811 |
| Shakespeare (noniid) | 25 | 17 | 12 | 13 | 30 | 20 | 57 | **10** | 11 |
| | 30 | 32 | 19 | 20 | 45 | 36 | 78 | **17** | 20 |
| | 35 | 61 | **27** | **27** | 85 | 68 | 177 | 28 | 48 |

as follows: FedSMU (10.43 seconds), FedAvg (10.15 seconds), FedEF-HS (10.46 seconds), FedLion (10.63 seconds), SCAFFOLD (10.38 seconds). Experiments demonstrate that in a single communication round, our algorithm introduces no significantly additional time overhead compared to other algorithms. Therefore, the results using wall-clock time as a metric are similar to those measured by communication rounds. We will not include a separate plot here and please refer to Figure 3 and Table 10.

# G COMPARISON WITH MORE ALGORITHMS

## G.1 COMPARISON WITH OTHER ABLATION ALGORITHMS

To verify the effectiveness of different FL algorithms built upon the Lion optimizer in terms of the generalization and compression performance, we design additional variants of FL incorporated with Lion, namely Fed-LocalLion and Fed-GlobalLion. Specifically, Fed-LocalLion executes the Lion optimizer locally in parallel at clients, with the server performing model and momentum aggregation via a weighted summation. On the other hand, Fed-GlobalLion conducts the vanilla SGD locally, treats model aggregation as a pseudo-gradient on the server side, and updates the global model through the Lion optimizer. See Appendix H for the detail of these two algorithms.

Our FedSMU consistently outperforms the other variants, as illustrated in Figure 4. It is worth noting that FedSMU also integrates additional model compression, whereas these variants require even more communication overhead than FedAvg. This suggests that our FedSMU design effectively harnesses the benefits of Lion, enhancing the generalization while compressing the communication load.

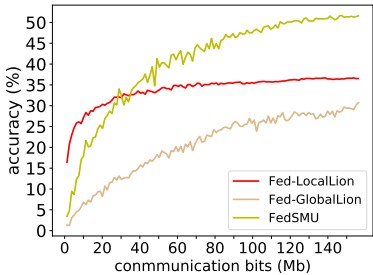

Figure 4: Convergence performance vs. number of communication bits on CIFAR-100 dataset and small CNN network, with 100 clients and 10% participation rates, Dirichlet-0.25 for different ablation algorithms and FedSMU.

## G.2 COMPARISON WITH SCAFFOLD (KARIMIREDDY ET AL., 2020)

SCAFFOLD (Karimireddy et al., 2020) aims to mitigate the client variance by designing and iteratively updating the control variates, which refers to the variance between the uploaded client updates. And thus it shows superior performance in most datasets.

Due to the performance gap between FedSMU and SCAFFOLD(Karimireddy et al., 2020) on CIFAR-10, we have considered whether FedSMU can outperform SCAFFOLD at the cost of increased communication overhead.

We conduct the experiment by increasing the number of rounds to 6000 and comparing FedSMUMC with SCAFFOLD on CIFAR-10 dataset, using a total of 100 clients with a partial participation ratio of 0.03 (represented in the Table 11 by L).

The experimental results are shown in the Table 11. It can be found that on the CIFAR-10 dataset with Dir (0.6)-L, FedSMU achieves a slightly higher accuracy compared SCAFFOLD when the number of communication rounds reaches 6000. We also find that when FedSMUMC is used at additional overhead of storing and communicating the momentum, it can perform better than SCAFFOLD over the same 4000 rounds.

However, it should be noted that there is a trade-off between the training time and test accuracy. A larger number of communication rounds leads to a slight performance improvement. For example, with FedSMU on CIFAR-10 with Dir (0.6)-L, an accuracy of 81.96% is achieved after 4000 rounds in the original manuscript, and an improvement of 0.52% is obtained with 2000 more rounds (6000 rounds in total), which exceeds that of SCAFFOLD. Therefore, we currently set 4000 rounds as the maximum communication round for all the algorithms considering the training efficiency.

Table 11: Performance comparison under different settings on CIFAR-10 dataset and small CNN network. Bold numbers indicate the best performance.

| | | | Top-1 Test Accuracy (%). | | |
|---|---|---|---|---|---|
| Dataset | Setting | Round | SCAFFOLD | FedSMU | FedSMUMC |
| CIFAR-10 | Dir(0.25)-L | 4000 | **81.6** | 79.66 | 80.47 |
| | | 6000 | **81.6** | 80.12 | 81.1 |
| | Dir(0.6)-L | 4000 | 82.36 | 81.96 | **82.67** |
| | | 6000 | 82.36 | 82.48 | **83.23** |

Besides, we have compared our FedSMU with SCAFFOLD on new MNIST dataset. Additionally, we also conducted another performance comparison scenario ($Dir = 0.6$) on CIFAR-100 using ResNet-18. The experimental results in Table 12 show that FedSMU outperforms SCAFFOLD on CIFAR-100. However, for the simpler MNIST dataset, FedSMU performs slightly worse than SCAFFOLD. This slight degradation on MNIST is likely due to the lower degree of heterogeneity in the dataset.

In a federated heterogeneous scenario with CIFAR-100, which consists of 100 categories compared to MNIST's 10 categories, each client typically handles a subset of 13-16 or 20-25 categories when $Dir = 0.25$ or $Dir = 0.6$, respectively. This high degree of heterogeneity in CIFAR-100 leads to greater deviations in model updates among clients, enabling FedSMU to achieve more significant improvements compared to MNIST. To further assess performance, we plan to explore a more complex dataset, Tiny ImageNet, for additional comparisons in future work.

Table 12: Performance comparison under different datasets, where L and H indicate low and high participation rates.

| | | | |
|---|---|---|---|
| | Top-1 Test Accuracy (%) . | | |
| Dataset (Model) | Setting | SCAFFOLD | FedSMU |
| MNIST on LeNet | Dir (0.6)-L | **98.4** | 97.37 |
| | Dir (0.6)-H | **98.2** | 97.47 |
| CIFAR100 on Resnet18 | Dir (0.6)-H | 53.90 | **54.25** |

### G.3 COMPARISON WITH DISTRIBUTED LION (LIU ET AL., 2024)

Here, we clarify the differences and advantages of our FedSMU compared to the D-Lion (Liu et al., 2024), as follows.

**1) Motivation.** Our FedSMU can simultaneously mitigate data heterogeneity and reduce communication compression through the symbolic operations. The analysis was carried out and verified by experiments (Figure 1). While D-Lion only consider to compress the communication.

**2) Scope of application.** Our FedSMU can deal with scenarios involving the partial client participation and multiple local updates, whereas D-Lion can not. Performing multiple local updates, in the federated settings, can effectively reduce the communication frequency and thus the overall traffic. Experimental results in Table 13 and Table 14 demonstrate that D-Lion fails in such scenarios with low client participation rates and multiple local updates, whereas FedSMU remains robust and performs well under these conditions.

**3) Algorithms design.** While both algorithms are based on the Lion optimizer, FedSMU fully leverages the structural advantages of the Lion optimizer, including weight decay in the global aggregation. In contrast, D-Lion primarily incorporates the momentum sliding averaging and symbolic operations at local update. This comprehensive utilization of the Lion optimizer structure may explain why the experimental performance of our FedSMU surpasses that of D-Lion.

**4) Compatibility with majority vote.** We have further extended FedSMU with majority vote, as FedSMU-MV. Experimental results show that FedSMU-MV achieves an accuracy of 47.66% on CIFAR-100, slightly lower than FedSMU's 51.79% under the same settings (number of clients = 100, fraction = 0.1, Dirichlet = 0.25). This indicates that majority vote is compatible with our algorithm. The slight accuracy drop may result from FedSMU's symbolic model updates. Applying majority vote to the 1-bit results could further suppress some clients' model update information due to the dominant update direction.

Below, we provide the details of the hyperparameters used in our experiments.

- To ensure a fair comparison, both algorithms are evaluated on the ClFAR-10 dataset, using non-llD data (Dirichlet distribution with a parameter of 0.25), with a total of 10 clients and a batch size of 50.
- For FedSMU and FedAvg, we adopt the same parameter settings as outlined in Appendix A.
- For D-Lion, we performed a grid search. The learning rate ($\epsilon$) is selected from {0.00005, 0.0005, 0.005, 0.015}, the weight decay ($\lambda$) is chosen from {0.0005, 0.005, 0.001, 0.01} and $\beta_1$ $\beta_2$ are selected from {0.9,0.99}. For Table 13, the selected values are $\epsilon = 0.0005$, $\lambda = 0.001$, $\beta_1 = 0.9$, $\beta_2 = 0.99$. For Table 14, the selected values are $\epsilon = 0.015$, $\lambda = 0.01$, $\beta_1 = 0.9$, $\beta_2 = 0.9$.

Table 13: Performance comparison on CIFAR-10 dataset with small CNN network, where F and P indicate full and partial participation rates, and K is the number of local updates.

| Setting | Algorithm | $K = 1$ with F | $K = 1$ with P | $K = 5$ with F | $K = 5$ with P |
|---------|-----------|------------|------------|------------|------------|
| | | Top-1 Test Accuracy (%) . | | | |
| Dir-0.25 | FedSMU | 32.47 | **38.35** | **77.97** | **75.14** |
| | D-Lion | 34.03 | 24.58 | 77.62 | 34.48 |
| | FedAvg | **79.64** | / | / | / |
| iid | FedSMU | 81.84 | **77.99** | **82.37** | **81.71** |
| | D-Lion | **82** | 29.06 | 82.36 | 44.05 |
| | FedAvg | 79.53 | / | / | / |

Table 14: Performance comparison on CIFAR-10 dataset with small CNN, where F and P indicate full and partial participation rates, and K is the number of local updates.

| Setting | Algorithm | $K = 100$ ($K' = 1$ epoch) with F | $K = 500$ ($K' = 5$ epochs) with F | $K = 500$ ($K' = 5$ epochs) with P |
|---------|-----------|----------|----------|----------|
| | | Top-1 Test Accuracy (%) . | | |
| Dir-0.25 | FedSMU | **82.24** | **82.0** | **82.08** |
| | D-Lion | 82.19 | 81.6 | 51.23 |

Specifically, from the result in Table 13 and Table 14, we have following observations.

- With full participation and one local update (i.e., $K = 1$ with F), FedSMU performs slightly worse than D-Lion. However, in scenarios with a partial participation, FedSMU consistently outperforms

D-Lion. This is intuitive, as D-Lion does not maintain a complete global model at the server and only aggregates the global model updates. Thus in the partial participation settings, asynchronous clients can only save a stale global model. As a result, these clients may receive the global model updates, which, however, cannot be leveraged to recover the exact global model of the current round.

- With multiple local updates (i.e., $K > 1$), FedSMU consistently outperforms D-Lion. This performance improvement can be attributed to the different approaches to weight decay. Specifically, the hyperparameter $\gamma_2$ (denoted as $\lambda$ in D-Lion) controls the weight decay (or $L_2$ penalty) coefficient. In FedSMU, the regularization is applied to the global model $x_t$, potentially mitigating overfitting and thus enhancing generalization. In contrast, D-Lion applies this regularization to the local model $x_{t-1}^i$. As a result, when the local updates occur multiple times, D-Lion's regularization primarily affects the local model, and does not directly improve the generalization capability of the global model. Consequently, when finally evaluating the generalization performance of the global model, FedSMU demonstrates a significant advantage over D-Lion.

- In heterogeneous scenarios, the performance of both FedSMU and D-Lion is poorer than that of FedAvg, especially when $k$ is small. This is an interesting and somewhat unexpected finding, which we speculate is due to the data heterogeneity. In the heterogeneous settings, each client samples a mini-batch of data for training and performs only a single time of update, followed by the application of the sign operation to the model update. Since the local update occurs only once, it introduces a substantial sampling variance and inter-client variance. The sign operation, which normalizes the magnitude of updates, may inadvertently amplify this variance between clients, leading to an unstable or even divergent global model aggregation.

## G.4 COMPARISON WITH EF21 (RICHTÁRIK ET AL., 2021)

We further explore Error Feedback 2021 (Richtárik et al., 2021) algorithm as a state-of-the-art method for Top-K compression and make a comparison with it.

Experiments on CIFAR-10 and CIFAR-100 are conducted. We set a total of 100 clients with different participation rate (0.03% and 0.1%, represented in Table 15 by L and H) and use Dirichlet-0.25. The experimental results are shown in Table 15.

On CIFAR-100, FedSMU still shows a high performance. While on CIFAR-10, the accuracy of FedSMU can be higher than EF21 with a lower participation. These results strongly demonstrate the superiority of FedSMU in complex image classification tasks, especially under a low client participation rate, which may result from the sign operation promoting the fair contribution of clients effectively to the global model update.

Table 15: Performance comparison under different datasets with small CNN network, where L and H indicate low and high participation rates. Bold numbers indicate the best performance.

| Dataset | Setting | FedSMU | EF21 |
|---------|---------|--------|------|
| | Top-1 Test Accuracy (%). | | |
| CIFAR-10 | Dir(0.25)-L | **79.66** | 74.43 |
| | Dir(0.25)-H | 80.67 | **81.51** |
| CIFAR-100 | Dir(0.25)-H | **51.79** | 50.07 |

## G.5 COMPARISON WITH ADAPTIVE ALGORITHMS

We compare with two adaptive algorithms in (Wang et al., 2022): the optimization-based FedAMS and the compression-based FedCAMS. FedAMS is designed to accelerate the convergence using momentum, while FedCAMS extends FedAMS by further compressing the upload communication. Experiments are conducted on CIFAR-10 and CIFAR-100 datasets. We use a total of 100 clients with a partial participation ratio of 0.1 and employ a Dirichlet distribution with a concentration parameter of 0.25. The experimental results are presented in the Table 16.

The experimental results demonstrate that on the CIFAR-10 dataset, FedSMU also outperforms FedCAMS but is slightly inferior to FedAMS, while FedSMU exhibits a superior performance compared to FedAMS and FedCAMS on the CIFAR-100 dataset. The results strongly demonstrate the superiority of FedSMU in complex image classification tasks, even comparable to the uncompressed federated adaptive algorithm, which may result from promoting the fair contribution of clients effectively to the global model update.

Table 16: Performance comparison under different datasets with 100 clients and 10% participation rates, Dirichlet-0.25, small CNN network. Bold numbers indicate the best performance.

| | Top-1 Test Accuracy (%) . | | |
|---|---|---|---|
| Dataset | FedSMU | FedAMS | FedCAMS |
| CIFAR-10 | 80.67 | **82.47** | 80.15 |
| CIFAR-100 | **51.79** | 47.97 | 48.3 |

### G.6 MAGNITUDE UNIFORMITY INDEX OF MORE ALGORITHMS

We provide Figure 5, Tables 17 and Table 18 showing the correlation between Magnitude Uniformity (MU), data heterogeneity, and accuracy of three different algorithms.

The results indicate that with FedAvg, data heterogeneity significantly amplifies the differences in the magnitude of model updates across clients, leading to unstable global aggregation and poorer generalization performance. While SCAFFOLD reduces variance to address these differences, FedSMU directly ensures consistency across all model updates through symbolic operations. Those two approaches enhances Magnitude Uniformity among clients, ultimately improving accuracy.

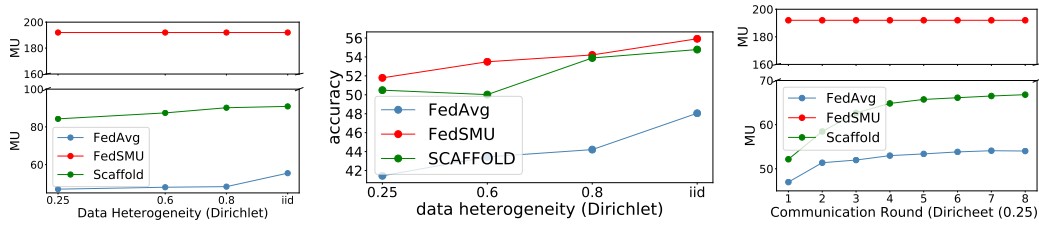

(a) MU index vs. heterogeneity        (b) Accuracy vs. heterogeneity        (c) MU vs. communication round

Figure 5: Magnitude uniformity (MU) index and top validation accuracy of FedAvg and FedSMU (ours) on CIFAR-100 with small CNN network.

Table 17: Accuracy (%) vs. data heterogeneity on CIFAR-100 with small CNN network, involving 100 clients with a participation rate of 10%.

| Algorithm | Dirichlet-0.25 | Dirichlet-0.6 | Dirichlet-0.8 | iid |
|---|---|---|---|---|
| FedSMU | 51.79 | 53.49 | 54.21 | 55.92 |
| FedAvg | 41.44 | 43.44 | 44.21 | 48.05 |
| SCAFFOLD | 50.49 | 50.02 | 53.89 | 54.78 |

Table 18: MU vs. data heterogeneity on CIFAR-100 with small CNN network, involving 100 clients with a participation rate of 10%.

| Algorithm | Dirichlet-0.25 | Dirichlet-0.6 | Dirichlet-0.8 | iid |
|---|---|---|---|---|
| FedSMU | 192.00 | 192.00 | 192.00 | 192.00 |
| FedAvg | 46.98 | 48.00 | 48.32 | 55.46 |
| SCAFFOLD | 84.25 | 87.41 | 90.15 | 90.86 |

# H OTHER ALGORITHMS

---

**Algorithm 2:** FedAvg

1 **Input:** $x_0, \eta_L^t$
2 **for** *each round $t = 1, 2, \ldots, T$* **do**
3    sample clients $\mathcal{P}_t \subseteq \mathcal{M}$
4    **for** *each client $i \in \mathcal{P}_t$ in parallel* **do**
5       receive and initialize local model $x_{i,0}^t = x_{t-1}$
6       **for** *each local step $k = 1, 2, \ldots, K$* **do**
7          $x_{t,k}^i = x_{t,k-1}^i - \eta_L^t(\nabla F_i(x_{t,k-1}^i, \xi_{t,k-1}^i))$
8       **end**
9       $\Delta x_t^i = x_{t,K}^i - x_{t,0}^i$ and send $\Delta x_t^i$ to server
10    **end**
11    **for** *each client $i \notin \mathcal{P}_t$ in parallel* **do**
12       $\Delta x_t^i = \Delta x_{t-1}^i, x_t^i = x_{t-1}^i$
13    **end**
14    // at server:
15       $x_t = x_{t-1} + \frac{1}{p}\sum_{i \in \mathcal{P}_t} \Delta x_t^i = \frac{1}{p}\sum_{i \in \mathcal{P}_t} x_t^i$
16       $\widetilde{x}_t = \frac{1}{m}\sum_{i \in \mathcal{M}} x_t^i$
17 **end**

---

FedEF is shown in Algorithm 3. At each round $t \in [T]$, a subset of clients $\mathcal{N}_t \subseteq \mathcal{M}$ are active, and the server transmits its current model $x_t$ to these clients. Each active client then performs local SGD (Line 8), sends the compressed model difference $\Delta_t^i$ back to the server (Lien 11) and updates the local error accumulator $e_t^i$ (Line 12). In Line 15, $\mathcal{C}$ denotes a compressor (e.g., TopK and Sign are two choices of compressors that correspond to FedEF-TopK and FedEF-Sign, respectively. The server aggregates the compressed model difference $\Delta_t^i$ to update $x_{t+1}$ (Line 15).

---

**Algorithm 3:** FedEF-SGD

1 **Server Initialization**: $x_0$;
2 **Client Initialization**: $e_0^i = 0$;
3 **for** *each round $t = 1, 2, \ldots T$* **do**
4    sample clients $\mathcal{N}_t \subseteq \mathcal{M}$
5    **for** *each client $i \in \mathcal{N}_t$ in parallel* **do**
6       receive and initialize local model $y_{t,0}^i = x_t$
7       **for** *each local step $k = 1, 2, \ldots, K$* **do**
8          $y_{t,k}^i = y_{t,k-1}^i - \eta \nabla F_i(y_{t,k-1}^i, \xi_{t,k-1}^i)$
9       **end**
10       $g_t^i = y_{t,K}^i - y_{t,0}^i$
11       send $\Delta_t^i = \mathcal{C}(g_t^i + e_t^i)$ to server
12       $e_t^i = e_t^i + g_t^i - \Delta_t^i$
13    **end**
14    // at server:
15    $x_{t+1} = x_t + \eta_g(\frac{1}{n}\sum_{i=1}^n \Delta_t^i)$
16    broadcast $x_{t+1}$
17 **end**

---

FedLion is shown in Algorithm 4. At each round $t \in [T]$, a subset of clients $\mathcal{N}_t \subseteq \mathcal{M}$ are active, and the server transmits its current model and momentum, $x_t, M_t$ to these clients. Each active client then performs local SGD to calculate the gratitude $g_{t,k}^i$ (Line 8) and uses the Lion optimizer to update local model. Inter-valued model difference $\frac{1}{\gamma_1}\Delta_t^i$ is sent to the server alone with the local momentum (Lines 13 and 14).

GlobalLion, as a variant evaluated in the ablation study of our FedSMU, is shown in Algorithm 5. At each round $t \in [T]$, a subset of clients $\mathcal{N}_t \subseteq \mathcal{M}$ are active, and the server transmits its current model $x_t$ to these clients.

**Algorithm 4:** FedLion

1 **Server Initialization:** $x_0, M_1 = 0$;
2 **for** *each round* $t = 1, 2, \ldots T$ **do**
3      sample clients $\mathcal{N}_t \subseteq \mathcal{M}$
4      **for** *each client* $i \in \mathcal{N}_t$ *in parallel* **do**
5          receive and initialize local model and momentum $y_{t,0}^i = x_t, m_{t,0}^i = M_t$
6          **for** *each local step* $k = 1, 2, \ldots, K$ **do**
7              $g_{t,k}^i = \nabla F_i(y_{t,k-1}^i, \xi_{t,k-1}^i)$
8              $u_{t,k} = sign(\beta_1 m_{t,k-1}^i + (1 - \beta_1) g_{t,k}^i)$
9              $m_{t,k}^i = \beta_2 m_{t,k-1}^i + (1 - \beta_2) g_{t,k}^i$
10              $y_{t,k}^i = y_{t,k-1}^i - \gamma_1 u_{t,k}^i$
11          **end**
12          $\Delta_t^i = \frac{y_{t,K}^i - y_{t,0}^i}{\gamma_1}$
13          send $\Delta_t^i, m_{t,K}^i$ to server
14      **end**
15      // at server:
16      $M_{t+1} = \frac{1}{n} \sum_{i=1}^n m_t^i$
17      $x_{t+1} = x_t + \gamma_1 (\frac{1}{n} \sum_{i=1}^n \Delta_t^i)$
18      broadcast $x_{t+1}$ and $M_{t+1}$
19 **end**

Each active client then updates the local model (Line 8) and sends the model difference $g_t^i$ to server. The server aggregates the $g_t^i$ as the global model difference $G_t$ (Line 14) and uses the Lion optimizer to update.

**Algorithm 5:** Fed-GlobalLion

1 **Server Initialization:** $x_0, M_0 = 0$;
2 **for** *each round* $t = 1, 2, \ldots T$ **do**
3      sample clients $\mathcal{N}_t \subseteq \mathcal{M}$
4      **for** *each client* $i \in \mathcal{N}_t$ *in parallel* **do**
5          receive and initialize local model $y_{t,0}^i = x_t$
6          **for** *each local step* $k = 1, 2, \ldots, K$ **do**
7              $y_{t,k}^i = y_{t,k-1}^i - \eta \nabla F_i(y_{t,k-1}^i, \xi_{t,k-1}^i)$
8          **end**
9          $g_t^i = y_{t,K}^i - y_{t,0}^i$
10          send $g_t^i$ to server
11      **end**
12      // at server:
13      $G_t = \frac{1}{n} \sum_{i=1}^n g_t^i$
14      $U_t = sign(\beta_1 M_{t-1} + (1 - \beta_1) G_t)$
15      $M_t = \beta_2 M_{t-1} + (1 - \beta_2) G_t$
16      $x_{t+1} = x_t + \gamma_1 (U_t - \gamma_2 x_t)$
17      broadcast $x_{t+1}$
18 **end**

LocalLion, as a variant evaluated in the ablation study of our FedSMU, is shown in Algorithm 6. At each round $t \in [T]$, a subset of clients $\mathcal{N}_t \subseteq \mathcal{M}$ are active, and the server transmits its current model $x_t$ to these clients. Each active client then performs SGD (Line 8) and uses the Lion optimizer to further update model. The server aggregates the local model difference $\Delta_t^i$ to compute $x_{t+1}$.

FedSMUM, as a variant evaluated in the ablation study of our FedSMU, is shown in Algorithm 7. At each round $t \in [T]$, the server transmits its current model, $x_t$ to all clients. A subset of clients $\mathcal{N}_t \subseteq \mathcal{M}$ are active, and they perform as FedSMU. Stale clients do not update their models, but calculate the model difference to approximate $g_t^j$ (Line 17) and update momentum $m_t^j$ (Line 18) without transmission.

**Algorithm 6:** Fed-LocalLion

1  **Server Initialization**: $x_0$;
2  **Client Initialization**: $m_0^i = 0$;
3  **for** *each round* $t = 1, 2, ...T$ **do**
4     sample clients $\mathcal{N}_t \subseteq \mathcal{M}$
5     **for** *each client* $i \in \mathcal{N}_t$ *in parallel* **do**
6         receive and initialize local model $y_{t,0}^i = x_t$
7         **for** *each local step* $k = 1, 2, \ldots, K$ **do**
8            $y_{t,k}^i = y_{t,k-1}^i - \eta \nabla F_i(y_{t,k-1}^i, \xi_{t,k-1}^i)$
9         **end**
10        $g_t^i = y_{t,K}^i - y_{t,0}^i$
11        $u_t^i = sign(\beta_1 m_{t-1}^i + (1 - \beta_1)g_t^i)$
12        $m_t^i = \beta_2 m_{t-1}^i + (1 - \beta_2)g_t^i$ (for $i \notin \mathcal{N}_t$, $m_t^i = m_{t-1}^i$)
13        $y_t^i = y_{t,K}^i + \gamma_1(u_t^i - \gamma_2 y_{t,K}^i)$
14        $\Delta_t^i = y_t^i - y_{t,0}^i$
15        send $\Delta_t^i$ to server
16     **end**
17     // at server:
18     $x_{t+1} = x_t + \eta_g(\frac{1}{n}\sum_{i=1}^n \Delta_t^i)$
19     broadcast $x_{t+1}$
20 **end**

**Algorithm 7:** FedSMUM

1  **Server Initialization**: $x_0$;
2  **Client Initialization**: $m_0^i = 0$;
3  **for** *each round* $t = 1, 2, ...T$ **do**
4     sample clients $\mathcal{N}_t \subseteq \mathcal{M}$
5     **for** *each client* $i \in \mathcal{N}_t$ *in parallel* **do**
6         receive and initialize local model $y_{t,0}^i = x_t$
7         **for** *each local step* $k = 1, 2, \ldots, K$ **do**
8            $y_{t,k}^i = y_{t,k-1}^i - \eta \nabla F_i(y_{t,k-1}^i, \xi_{t,k-1}^i)$
9         **end**
10        $g_t^i = y_{t,K}^i - y_{t,0}^i$
11        $u_t^i = sign(\beta_1 m_{t-1}^i + (1 - \beta_1)g_t^i)$
12        $m_t^i = \beta_2 m_{t-1}^i + (1 - \beta_2)g_t^i$
13        send $u_t^i$ to server
14     **end**
15     **for** *each client* $j \notin \mathcal{N}_t$ *in parallel* **do**
16         receive and initialize local model $y_{t,K}^j = x_t$
17         $g_t^j = y_{t,K}^j - (1 - \gamma_1\gamma_2)y_{t-1,K}^j$
18         $m_t^j = \beta_2 m_{t-1}^j + (1 - \beta_2)\gamma_1 g_t^j$
19     **end**
20     // at server:
21     $x_{t+1} = x_t + \gamma_1(\frac{1}{n}\sum_{i=1}^n u_t^i - \gamma_2 x_t)$
22     broadcast $x_{t+1}$
23 **end**

FedSMUMC, as a variant evaluated in the ablation study of our FedSMU, is shown in Algorithm 8. The basic procedure is equivalent to FedSMU. At each round $t \in [T]$, a subset of clients $\mathcal{N}_t \subseteq \mathcal{M}$ are active, and the server transmits its current model $x_t$ and global momentum $M_t$ to these clients. Local clients also additionally transfer $m_t^i$ back to the server (Line 13) and average them to update the momentum for the next round (Line 16).

**Algorithm 8:** FedSMUMC

**1 Server Initialization**: $x_0, M_0$;

**2 Client Initialization**: $m_0^i = 0$;

**3 for** *each round $t = 1, 2, \ldots T$* **do**

4      sample clients $\mathcal{N}_t \subseteq \mathcal{M}$

5      **for** *each client $i \in \mathcal{N}_t$ in parallel* **do**

6          receive and initialize local model $y_{t,0}^i = x_t$

7          **for** *each local step $k = 1, 2, \ldots, K$* **do**

8              $y_{t,k}^i = y_{t,k-1}^i - \eta \nabla F_i(y_{t,k-1}^i, \xi_{t,k-1}^i)$

9          **end**

10         $g_t^i = y_{t,K}^i - y_{t,0}^i$

11         $u_t^i = sign(\beta_1 M_t + (1 - \beta_1)g_t^i)$

12         $m_t^i = \beta_2 M_t + (1 - \beta_2)g_t^i$

13         send $u_t^i, m_t^i$ to server

14      **end**

15      // at server:

16      $M_{t+1} = \frac{1}{n} \sum_{i=1}^n m_t^i$

17      $x_{t+1} = x_t + \gamma_1(\frac{1}{n} \sum_{i=1}^n u_t^i - \gamma_2 x_t)$

18      broadcast $x_{t+1}, M_{t+1}$

**19 end**

**Algorithm 9:** Lion algorithm.

**1 Initialization**: $x_0, m_0$;

**2 for** *each round $t = 1, 2, \ldots T$* **do**

3      $g_t = \nabla f(x_t, \xi_t)$

4      $v_t = \beta_1 m_t + (1 - \beta_1)g_t$

5      $m_t = \beta_2 m_t + (1 - \beta_2)g_t$

6      $x_{t+1} = x_t + \gamma_1(\textbf{sign}(v_t) - \gamma_2 x_t)$

**7 end**

