# OpenReview forum: "FedSMU: Communication-Efficient and Generalization-Enhanced Federated Learning through Symbolic Model Updates"
_ICLR.cc/2025/Conference — Submitted to ICLR 2025_

### Official Review · Reviewer_38dT · 2024-11-03

**Soundness:** 2
**Presentation:** 1
**Contribution:** 2
**Rating:** 5
**Confidence:** 4

**Summary:**

This paper presents a new FL algorithm, FedSMU, that leverages a 1-bit compression and a weight decay strategy with the goal of achieving both communication reduction and improved generalization performance. The proposed algorithm leverages the concept of the Lion optimizer, along with a convergence analysis. The paper also presents numerical experiments of FedSMU, comparing its performance with various FL algorithms by varying degrees of client participation ratios and data heterogeneity using CIFAR-10, CIFAR-100, and Shakespeare datasets, with CNN and RNN models.

**Strengths:**

- The paper introduces an interesting metric "Magnitude Uniformity" to measure heterogeneity in training, which is effectively used to motivate the proposed method through the numerical results.
- The proposed method integrates the Lion optimizer to enhance communication efficiency, showing significant improvements over FedLion, which is based on a similar concept.
- The proposed method demonstrates superior performance compared to other methods, particularly under low participation rates (Table 4).
- The paper provides clear empirical demonstrations of the impact of client participation rates and data heterogeneity on the generalization performance of FedSMU.

**Weaknesses:**

- The convergence result is dependent on the upper bound of the gradients, which is not the case, for example, in the convergence result from FedLion. Moreover, the justification for Assumption 4.3 is weak or even irrelevant to the context of sign-based methods.
- The theoretical and experimental results seem disconnected. As discussed in Section 4, the convergence may be affected by the model dimension and the maximum participation intervals, which are not demonstrated or discussed in the experimental section. Instead, the experimental section focuses only on demonstrating the performance of FedSMU.
- The experimental results are difficult to compare across different settings. For instance, the experiments with CNN used client participation rates of 10\% and 3\% with 100 clients, whereas those with ResNet18 used 30\% and 10\% with 10 clients. The impact of model sizes could be demonstrated more effectively if consistent experimental settings were used for CNN and ResNet18, as shown in Tables 2 and 3.
- The definition of generalization and its corresponding performance metric are not clearly defined.
- Fed-LocalLion and Fed-GlobalLion algorithms are arbitrarily devised by the authors, and the ablation studies with these algorithms do not seem to provide meaningful insights into the proposed method.

**Questions:**

1. What is the definition of generalization performance?
1. Regarding Remark 4.6, FedLion seems to provide the relevant convergence rate for the Lion optimizer, even under weaker assumptions. Please confirm that.
1. The large model used a 30\% participation rate, whereas the other experiments used 10\% and 3\%. Could you clarify why these inconsistent settings were used? This inconsistency makes it challenging to compare the large model's performance with the smaller models. Please consider using the consistent experiment settings.
1. The performance of FedSMU with the large model appears marginal compared to SCAFFOLD. Therefore, the last sentence of Section 5.2.1 may not be accurate. Please confirm and clarify.
1. Could the authors include plots of SCAFFOLD in Figure 2? This may help illustrate the trade-off between performance and communication efficiency more effectively.
1. Which model was used in Section 5.2.2? Please specify for clarity.
1. Why was FedAvg chosen in Section 5.2.3 for benchmarking on data heterogeneity? Would SCAFFOLD not have been a more appropriate comparison?
1. The performance reported in Table 4 is very intriguing. Could you elaborate on why FedSMU works well with low participation rates?
1. The performance of FedLion (Table 4) also appears consistent across different participation rates. Could you explain these results? Are they possibly related to Lion optimizer?
1. There are duplicate references for Chen et al. 2024a,b. Please correct this duplication.
1. Typo: "smay till" should be corrected.
1. In Algorithm 4, the client initialization of $e_0^i$ does not seem necessary. Instead, $M_t$ should be initialized.
1. In Algorithm 5, the client initialization of $m_0^i$ does not seem necessary. Instead, $M_t$ should be initialized. The superscript $i$ in $M_{t-1}^i, M_t^i, G_t^i, U_t^i$ may be typos.

---

> ### Author Response · Authors · 2024-11-22
> **Response to Comments 1 and 2**
>
> We would like to thank the reviewer for the comments. In the following, we have provided our detailed responses to each of these comments.
>
> > Comment 1: The  Assumption 4.3 is weak or even irrelevant to the context of sign-based methods.
>
> **Response**:
>
> We would like to acknowledge that this is a stronger assumption, ensuring that both the compressed targets and the momentum term (the moving averages of gradients) in our theoretical analysis are bounded. Moreover, this assumption has been adopted in other federated optimization and sign-based compression studies [D1, D2]. Specifically, in [D1], the authors demonstrate the convergence of distributed SIGNSGD with momentum under Assumption 4, which is also the bounded gradient assumption.
>
> In contrast, while FedLion does not rely on this stronger assumption, it also introduces an additional stronger assumption (A.4 in the original manuscript of FedLion):
>
> ``There exists a constant **$\alpha\leq\frac{1}{3}$**, such that $\left\|\nabla f(x)-\nabla f_i(x)\right\|_1 \leq \alpha\|\nabla f(x)\|_1, \forall i, x .$''
>
> And we will clarify these points in the revised manuscript.
>
> [D1] Tao Sun, Qingsong Wang, Dongsheng Li, and Bao Wang. ``Momentum ensures convergence of signsgd under weaker assumptions.'' In *International Conference on Machine Learning*, pp. 33077–33099. PMLR, 2023.
>
> [D2] Sashank Reddi, Zachary Charles, Manzil Zaheer, Zachary Garrett, Keith Rush, Jakub Konecny, Sanjiv Kumar, and H Brendan McMahan. ``Adaptive federated optimization.'' *arXiv preprint arXiv:2003.00295* , 2020.
>
>
> > Comment 2: The theoretical and experimental results seem disconnected.
>
> **Response:**
>
> Thank you very much for your suggestion. We have done further analysis on the influence of model dimension and client participation rate on convergence speed in this reponse. This theoretical relationship is consistent with the conclusion made in other papers [D3].
>
> Theoretically, a lower client participation rate (i.e., a larger $\tau_{max}$) leads to a slower algorithm convergence. Similarly, a higher model dimension $d$ also results in a slower algorithm convergence. To validate this, we have conducted the following experiments.
>
> All of these experiments are done on CIFAR-10 dataset with Dirichlet-0.25. To illustrate the relationship between the model dimension and convergence rate, we use two different CNN models ($d1=797248$ and $d2=1723648$) to study the impact of model dimension. Note that here we modify the size of the convolutional layers, keeping the model depth constant. The number of clients is 100 with the participation ratio of 0.1. To illustrate the relationship between participation rate and convergence rate, We use the CNN network with $d1=797248$ and set the number of clients as 100 with different participation ratio ($\frac{n}{m}=0.03$ and $\frac{n}{m}=0.1$, represented in the table by L and H) to demonstrate the influence of client participation rate.
>
> The experimental results in Table D1 show that when the participation rate is higher (i.e., the $\tau_{max}$ is smaller) and the dimension is smaller, the convergence speed can be faster, which matches with the Theory 4.4 and is intuitional.
>
> [D3] McMahan, Brendan, et al. ``Communication-efficient learning of deep networks from decentralized data.'' In *Artificial intelligence and statistics*. PMLR, 2017.
>
> **Table D1: Number of communication rounds to achieve test accuracy. Each table entry gives the number of communication rounds necessary to achieve the test accuracy. d1 and d2 indicate small and large dimensions, while L and H indicate low and high participation rates. Bold numbers indicate the smallest rounds.**
>
> | **Test Accuracy (%)** | **d1, H** | **d2, H** | **d1, L** |
> |-----------------------|-----------|-----------|-----------|
> | 40                   | **26**    | 30        | 46        |
> | 45                   | **33**    | 35        | 112       |
> | 50                   | 50        | **43**    | 193       |
> | 55                   | **65**    | 65        | 226       |
> | 60                   | 118       | **111**   | 384       |
> | 65                   | 288       | **275**   | 899       |
> | 67.5                 | **399**   | 409       | 949       |
> | 69                   | **507**   | 507       | 1025      |
> | 72.3                 | **642**   | 769       | 1752      |
> | 75                   | **1142**  | 1185      | 2221      |
> | 77.5                 | 1746      | **1745**  | 2878      |

---

> ### Author Response · Authors · 2024-11-22
> **Response to Comments 3 to 5 and Question 1**
>
> >Comment 3: The experimental results are difficult to compare across different settings.
>
> **Response:**
>
> In most federated learning scenarios, the total number of clients is set to 100 with the participation rate of 0.1, which is a classical experimental setting, like what FedLion and FedAvg do. Therefore, with the simple CNN network, we set 100 clients with participation rates of 0.1 and 0.03 to verify the performance of our algorithm.
>
> However, our computing resources (i.e., GEFORCE GTX 1080 Ti) are insufficient to support us to set 100 clients on the ResNet18 network, we thus can only select 10 clients and set the participation rate to 0.3.
>
> To illustrate the relationship between model dimension and convergence rate, we have conducted additional experiments, as detailed in our response to Comment 2.
>
> >Comment 4: The definition of generalization and its corresponding performance metric are not clearly defined.
>
> **Response:**
>
> We appreciate this comment and will further clarify the definition of generalization in the revised manuscript. Specifically, generalization refers to an algorithm's ability to achieve top test accuracy, where the test dataset is different from the training dataset.
>
> Furthermore, we consider an additional perspective on generalization to further evaluate the performance of our FedSMU algorithm. Here, generalization refers to a model's ability to achieve test performance at similar training error levels. Based on this definition, we compare the validation performance at similar training accuracy levels. The results in Table D2 show that on the CIFAR-10 and CIFAR-100 datasets, the FedSMU algorithm achieves the highest test accuracy and demonstrates the best generalization performance.
>
> **Table D2: Generalization performance comparison under various datasets.  Each table entry gives the average test accuracy on different training accuracy levels. "/" means it cannot reach the training accuracy. Bold numbers indicate the best performance.**
>
> | **Dataset**   | **Training Accuracy** | **FedAvg** | **SCAFFOLD** | **SCALLION** | **FedEF-HS** | **FedEF-TopK** | **FedEF-Sign** | **FedLion** | **FedSMU** |
> |---------------|------------------------|------------|--------------|--------------|---------------|----------------|---------------|-------------|------------|
> | **CIFAR-10**  | 83-84                 | 75.14     | **77.49**    | 75.9         | 75.63         | 75.48          | 75.7          | 77.22       | 77.1       |
> |               | 85-86                 | 76.17      | 76.37        | 76.87        | 76.83         | 76.83          | 77.4          | 78.3        | **78.39**  |
> |               | 87-88                 | 78.56      | 79.24        | 77.79        | 77.85         | 77.74          | /             | 79.23       | **79.78**  |
> | **CIFAR-100** | 66-67                 | 40.08      | 45.76        | 40.56        | /             | 41.28          | /             | 45.55       | **49.03**  |
> |               | 68-69                 | 40.6       | 46.16        | 40.88        | /             | 41.81          | /             | 46.15       | **50.04**  |
> |               | 70-71                 | 40.9       | 46.76        | 41.23        | /             | 42.34          | /             | 46.56       | **50.85**  |
>
> >Comment 5: Fed-LocalLion and Fed-GlobalLion algorithms do not seem to provide meaningful insights into the proposed method.
>
> **Response:**
>
> Please note that we implemented these two variant algorithms in the ablation studies, and compared them with FedSMU to verify our motivation. Specifically, the symbolization of local client updates, which is introduced to balance the contribution of each client and prevent dominance by relatively large update values, significantly improves the overall generalization performance. Therefore, we designed Fed-LocalLion and Fed-GlobalLion to execute the Lion optimizer on the client and server sides, respectively. In these setups, the uploaded model updates maintain full precision rather than using the symbolization.
>
> The comparison results, as shown in Figure 3, clearly illustrate that FedSMU outperforms both Fed-LocalLion and Fed-GlobalLion. This highlights that the improvement stems not solely from a direct application of the Lion optimizer in the federated settings, but also from the deliberately designed symbolization of all the client updates.
>
> > Question 1: What is the definition of generalization performance?
>
> **Response:**
>
>  Please refer to our response to Comment 4.

---

> > ### Comment · Reviewer_38dT · 2024-11-26
> >
> > I appreciate the authors’ detailed responses to each comment and question. They have addressed and clarified most of my comments/questions. I believe this paper/revision will contain several results that could potentially strengthen the paper, which need careful editing to ensure effective organization and demonstration.
> >
> > However, I believe the paragraph comparing Fed-LocalLion and Fed-GlobalLion would be better suited for the Appendix. Please consider that. The comparisons appear trivial, and the outcomes are expected, which could potentially lead to unnecessary confusion.

---

> > > ### Author Response · Authors · 2024-11-28
> > > **Official Comment by Authors**
> > >
> > > Dear Reviewer 38dT:
> > >
> > > We sincerely appreciate your time and effort in reviewing our paper and providing these valuable comments. We have incorporated these clarifications and new experiments into the revised manuscript that was newly uploaded. Specifically, we have moved the paragraph comparing our proposed FedSMU with Fed-LocalLion and Fed-GlobalLion in Appendix G.1 of the revised manuscript.
> > >
> > > For the summary of our changes in the revised manuscript, please refer to our General Response. We believe that these insights will further contribute to enhancing the quality of the final version of our manuscript.

---

> ### Author Response · Authors · 2024-11-22
> **Response to Questions 2 to 13**
>
> >Question 2: FedLion seems to provide the relevant convergence rate for the Lion optimizer, even under weaker assumptions.
>
> **Response:**
>
> Please note that FedLion cannot be reduced to a standard Lion optimizer, because it merely parallelizes the execution of the Lion optimizer on the client side and incorporates multi-precision quantization for communication compression. Therefore, even when $K=1$ and $m=1$, it does not degrade to a standard Lion optimizer, and its convergence rate cannot be directly applied to the Lion optimizer.
>
> In contrast, when $K=1$ and $m=1$ in our algorithm, it degenerates to the standard Lion optimizer, and its convergence rate can be applied to the Lion optimizer.
>
> >Question 3: The large model used a $30\%$ participation rate, whereas the other experiments used $10\%$ and $3\%$. Could you clarify why these inconsistent settings were used?
>
> **Response:**
>
>  Please refer to the our response to Comment 3.
>
> >Question 4: The last sentence of Section 5.2.1 may not be accurate.
>
> **Response:**
>
> We will clarify this in the revised manuscript, as follows.
>
> The results indicate that FedSMU demonstrates a strong performance on ResNet18 for both the CIFAR-10 and CIFAR-100 datasets, outperforming most baseline methods, though its accuracy remains slightly below Scaffold on the CIFAR-10 dataset.
>
> >Question 5: Could the authors include plots of SCAFFOLD in Figure 2?
>
> **Response:**
>
> Thanks for the suggestion and we will incorporate this in the revised manuscript.
>
> >Question 6: Which model was used in Section 5.2.2?
>
> **Response:**
>
> We used the small CNN network in Section 5.2.2, and we introduced its specific settings in Appendix A of the original manuscript.
>
> >Question 7: Why was FedAvg chosen in Section 5.2.3 for benchmarking on data heterogeneity? Would SCAFFOLD not have been a more appropriate comparison?
>
> **Response:**
>
> We have incorporated the experimental results of SCAFFOLD into Table 5 and show it in Table D3 here. The results continue to demonstrate that FedSMU is less impacted by the degree of data heterogeneity and achieves a high accuracy on the CIFAR-100 data set.
>
> **Table D3: Performance comparison under different datasets**
>
> | **Algorithm** | **Dirichlet-0.25** | **Dirichlet-0.6** | **Dirichlet-0.8** | **iid**   |
> |---------------|--------------------|-------------------|-------------------|-----------|
> | **FedSMU**    | 51.79             | 53.49            | 54.21            | 55.92     |
> | **FedAvg**    | 41.44             | 43.44            | 44.21            | 48.05     |
> | **SCAFFOLD**  | 50.49             | 50.02            | 53.89            | 54.78     |
>
> >Question 8:  Could you elaborate on why FedSMU works well with low participation rates?
>
> **Response:**
>
> We would like to attribute this to the use of symbolic operations for client updates, which effectively leverages each client's update even at very low participation rates. Specifically, when the client participation rate is low, data heterogeneity may cause the update of certain clients to dominate due to larger magnitudes. Symbolic operations can mitigate this by normalizing the update amplitudes, ensuring that the contributions of all clients are fully considered.
>
> >Question 9: Could you explain these results? Are they possibly related to Lion optimizer?
>
> **Response:**
>
>  We speculate that the primary reason lies in the symbolic operation of model updates in the Lion optimizer, as detailed in our response to Q8. These symbolic operations mitigate the impact of data heterogeneity by normalizing update amplitudes, ensuring that all clients' contributions are fully considered. Whether other structures of the Lion optimizer also play a significant role in mitigating the influence of data heterogeneity remains an open question and an interesting direction for future research.
>
> >Question 10: There are duplicate references for Chen et al. 2024a,b.
>
> **Response:**
>
> Thanks for your correction, and we will delete the duplicate reference in the revised manuscript.
>
> >Question 11: Typo: ``smay till'' should be corrected.
>
> **Response**
>
> Thank you for your correction, and we will correct it to "may still'' in the revised manuscript.
>
> >Question 12: In Algorithm 4, the client initialization of $e_0^i$ does not seem necessary. Instead, $M_t$ should be initialized.
>
> **Response:**
>
> Thank you for your correction, and we will correct it to initialize $M_{1}=0$ on the server side.
>
> >Question 13: In Algorithm 5, the client initialization of $m_0^i$ does not seem necessary. Instead, $M_t$ should be initialized. The superscript $i$ in $M_{t-1}^i$,$M_t^i$,$G_t^i$,$U_t^i$ may be typos.
>
> **Response:**
>
> Thank you for your correction, and we will correct it to initialize $M_{0}=0$ on the server side. The superscript $i$ in $M_{t-1}^i$,$M_t^i$,$G_t^i$,$U_t^i$ are indeed typos and we will delete it in the revised manuscript.

---

### Official Review · Reviewer_HTgR · 2024-11-03

**Soundness:** 3
**Presentation:** 3
**Contribution:** 2
**Rating:** 5
**Confidence:** 4

**Summary:**

This paper proposed the FedSMU algorithm to reduce communication costs in federated learning optimization problems. By symbolizing local client updates, FedSMU balanced client contributions and prevented dominance by large updates for Lion optimizer. Theoretical analysis and experiments showed comparable convergence and better accuracy.

**Strengths:**

The paper addresses an important problem in FL and is easy to follow. The empirical results validate the effectiveness of the proposed method, and the authors also provide a theoretical analysis to support it.

**Weaknesses:**

* I believe some descriptions in the paper need to be refined for greater precision. For instance, in Section 3, the statement, “when the number of local update steps is set to  $K=1$ , the entire federated learning process reduces to the standard Lion algorithm,” is somewhat misleading. Even with $K=1$ , there is still a slight mismatch between the standard Lion algorithm and the proposed FedSMU.

* Remark 4.5 is somewhat confusing. The authors state, "we use a 1-bit quantization compression method. If a higher-bit compression (e.g., $\alpha$-bit) is used, the additional coefficient α will further slow down the overall convergence rate." This conclusion seems non-intuitive and not straightforward, as there is no clear indication in the theorem of how the convergence rate is influenced by $\alpha$. Typically, using higher-bit compression results in less compression loss, which would suggest improved performance rather than a slower convergence rate.

* It is unclear to me why $\tau_{\max}$ is used as an indicator for studying partial participation settings. Could you provide the practical value of $\tau_{\max}$ used in your experiments? Is there a specific reason for choosing this $\tau_{\max}$ indicator instead of the more common “sampling ratio $n/m$” (sampling $n$ out of $m$ clients), aside from practical considerations? It seems that in the experimental section, the participation ratio is also used.

* Given that Lion is an adaptive optimization strategy, it is important to discuss and compare it with other adaptive optimization methods, particularly those focused on communication efficiency, such as [1] or communication-efficient versions of AdamW in FL.

[1] Wang, Y., Lin, L., and Chen, J. Communication-efficient adaptive federated learning.

**Questions:**

* The convergence rate appears to be unrelated to $\gamma_2$. After briefly reviewing the proof in the appendix, it seems that the authors assume  $|\gamma_2 x_t^j| \leq 1$. This is an uncommon assumption in FL optimization (or even in general optimization). Could you provide more details or clarification regarding this assumption?

* Some formatting could be improved: for instance, the scaling of Figure 1, the presentation of $\min$ in equation 1, and the $\min$ and $\max$ in Theorem 4.4.
* It would be helpful to specify the model (e.g., CNN or ResNet) in the captions or when describing figures/tables, as it can be unclear which model was used for the results.

---

> ### Author Response · Authors · 2024-11-22
> **Response to Comments 1 to 3**
>
> We would like to thank the reviewer for the comments. In the following, we have provided our detailed responses to each of these comments.
>
> >Comment 1: In Section 3, the statement, “when the number of local update steps is set to $K=1$, the entire federated learning process reduces to the standard Lion algorithm,” is somewhat misleading.
>
> **Response:**
>
> We would like to thank the reviewer for pointing out this issue. A more precise expression would be: “when the number of local update steps and the number of clients are set to $K=1$ and $m=1$, respectively, the entire federated learning process reduces to the standard Lion algorithm.” This will be corrected in the revised manuscript.
>
> >Comment 2: Remark 4.5 is somewhat confusing.
>
> **Response:**
>
> First, we would like to acknowledge that for the general quantization-based compression algorithms, a higher precision quantization may often lead to a faster convergence. However, this may not hold for our FedSMU. This conclusion is based on the Lion optimizer and our previous analysis, as follows.
>
> **1)** The original Lion manuscript demonstrated that 1-bit quantization (via the sign operation) in centralized learning enhances the algorithm's convergence and generalization compared to other optimization techniques. That is, the authors analyzed that the sign operation introduces noise into the updates, which serves as a form of regularization, thereby improving both convergence and generalization.
>
> **2)** We also observe that in the federated learning setting, symbolizing client model updates (i.e., normalizing the magnitude of each model parameter at local clients) mitigates the model heterogeneity arising from data heterogeneity. This, in turn, enhances the overall performance of the globally aggregated model at the server.
>
> Consequently, we conclude that in our optimized structure, 1-bit quantization outperforms higher-bit or even full-bit quantization, since multi-bit compression does not guarantee that the update amplitude of each client is consistent. Experimental results in Table C1 further validate that for our designed optimization algorithm, using a higher-bit compression does not enhance the algorithm's convergence or generalization.
>
> **Table C1: Number of communication rounds to achieve test accuracy. Each table entry gives the number of rounds of communication necessary to achieve the test accuracy. Bold numbers indicate the smallest rounds.**
>
> | **Test Accuracy (%)** | **1-bit (FedSMU)** | **3-bit** | **8-bit** |
> |-----------------------|-------------------|-----------|-----------|
> | 40                   | **26**           | 65        | 54        |
> | 45                   | **33**           | 91        | 68        |
> | 50                   | **50**           | 168       | 119       |
> | 55                   | **65**           | 285       | 185       |
> | 60                   | **118**          | 456       | 224       |
> | 65                   | **288**          | 833       | 342       |
> | 67.5                 | **399**          | 1260      | 401       |
> | 69                   | 507              | 1967      | **456**   |
> | 72.3                 | **642**          | /         | 643       |
> | 75                   | 1142             | /         | **1040**  |
> | 77.5                 | **1746**         | /         | 1979      |
>
> >Comment 3: It is unclear to me why $\tau_{max}$ is used as an indicator for studying partial participation settings.
>
> **Response:**
>
> Please note that $\tau_{max}$ represents the maximum participation interval among all clients across the communication rounds. While it is true that $\mathbb{E}(\tau_{max})=\frac{m}{n}$ for scenarios with uniformly random participation and constant participation rates per round, we would like to consider a more general scenario here. In cases of non-uniform random participation or varying participation rates across rounds, $\mathbb{E}(\tau_{max})$ may not equal to $\frac{m}{n}$. Therefore, it is more appropriate to use $\tau_{max}$ in this context.

---

> ### Author Response · Authors · 2024-11-22
> **Response to Comment 4 and Questions 1 to 3**
>
> >Comment 4: Given that Lion is an adaptive optimization strategy, it is important to discuss and compare it with other adaptive optimization methods, particularly those focused on communication efficiency.
>
> **Response:**
>
> Thank you for this suggestion. We have compared with the two adaptive algorithms as presented in the suggested literature [C1]: the optimization-based FedAMS and the compression-based FedCAMS. FedAMS is designed to accelerate the convergence using momentum, while FedCAMS extends FedAMS by further compressing the upload communication. Due to time limitation, our experiments are conducted only on CIFAR-10 and CIFAR-100 datasets. We use a total of 100 clients with a partial participation ratio of 0.1 and employ a Dirichlet distribution with a concentration parameter of 0.25. The experimental results are presented in the Table C2.
>
> The experimental results demonstrate that on the CIFAR-10 dataset, FedSMU also outperforms FedCAMS but is slightly inferior to FedAMS, while FedSMU exhibits a superior performance compared to FedAMS and FedCAMS on the CIFAR-100 dataset. The results strongly demonstrate the superiority of FedSMU in complex image classification tasks, even comparable to  the uncompressed federated adaptive algorithm, which may result from promoting the fair contribution of clients effectively to the global model update.
>
> [C1] Wang, Yujia, Lu Lin, and Jinghui Chen. ``Communication-efficient adaptive federated learning.'' In *International Conference on Machine Learning*. PMLR, 2022.
>
> **Table C2: Performance comparison under different datasets. Bold numbers indicate the best performance.**
>
> | **Dataset**   | **FedSMU** | **FedAMS**   | **FedCAMS** |
> |---------------|------------|--------------|-------------|
> | **CIFAR-10**  | 80.67      | **82.47**    | 80.15       |
> | **CIFAR-100** | **51.79**  | 47.97        | 48.3        |
>
> >Question 1: Authors assume $|\gamma_2 x_t^j\leq 1|$. This is an uncommon assumption in FL optimization (or even in general optimization). Could you provide more details or clarification regarding this assumption?
>
> **Response:**
>
> Though this is indeed a less common assumption, it has been demonstrated in literature [C2] (Equation 5 on Page 3). Literature [C2] clarifies in Abstract that ``Lion is a theoretically novel and principled approach for minimizing a general loss function f(x) while enforcing a bound constraint $||x||_{\infty}\leq \frac{1}{\gamma_2}$.'' Here $\gamma_2$ is the weight decay coefficient and $x_t$ is the model. Such an assumption has also been used in another algorithm based on Lion optimizer [C3] (Equation 7 on Page 4).
>
> [C2] Chen, Lizhang, et al. "Lion secretly solves constrained optimization: As lyapunov predicts.'' ICLR, 2024.
>
> [C3] Liu, Bo, et al. "Communication Efficient Distributed Training with Distributed Lion.'' *arXiv preprint arXiv:2404.00438* (2024).
>
> >Question 2: Some formatting could be improved: for instance, the scaling of Figure 1, the presentation of $min$ in equation 1, and the $min$ and $max$ in Theorem 4.4.
>
> **Response:**
>
> Thank you for pointing these formatting issues. We will modify them in the revised manuscript, including adjusting the scaling of Figure 1, and correcting the presentation of $\min$ and $\max$.
>
> >Question 3: It would be helpful to specify the model (e.g., CNN or ResNet) in the captions or when describing figures/tables.
>
> **Response:**
>
> We appreciate this suggestion, and will specify the model used for each figure and table in the revised manuscript. Specifically, For Table 3, we use the ResNet18 network. In the rest of the tables and figures, we use the simple CNN network (i.e., Letnet).

---

> > ### Comment · Reviewer_HTgR · 2024-11-26
> > **Thank the authors for their detailed rebuttal**
> >
> > Firstly, I thank the authors for their detailed rebuttal. I believe the authors have addressed most of my concerns, and I strongly encourage them to revise the paper accordingly and incorporate the new discussions into the camera-ready version.
> >
> > However, I still have some concerns regarding Remark 4.5 (from my original comment 2). While I acknowledge the authors’ ablation studies showing that relaxing communication bits in the FedSMU method does not improve results, I remain concerned about the claim: “We also observe that in the federated learning setting, symbolizing client model updates (i.e., normalizing the magnitude of each model parameter at local clients) mitigates the model heterogeneity arising from data heterogeneity.” This observation seems to stem from the specific design of FedSMU, which is inspired by the Lion optimizer that employs 1-bit quantization (via the sign operation). Therefore, I am concerned that the performance gain from using 1-bit quantization may be tied to the specific design of FedSMU rather than being generalizable to all FL algorithms.

---

> > > ### Author Response · Authors · 2024-11-28
> > > **Response to `` concerns regarding Remark 4.5 ''**
> > >
> > > Dear Reviewer HTgR:
> > >
> > > We are greatly encouraged by your general response stating that your concerns have been mostly addressed. We have incorporated these clarifications and new experiments into the revised manuscript that was newly uploaded. For the summary of our changes in the revised manuscript, please refer to our General Response. We believed that these insights will further contribute to enhancing the quality of the final version of our manuscript.
> > >
> > > On the other hand, for your concerns regarding Remark 4.5, we agree with your statement. The effect of the sign operation, which reduces the model heterogeneity and then improves the generalization performance, is currently only observed in the Lion architecture within the federated learning context.
> > >
> > > First, both experimentally (as shown in Figure 1 of the original manuscript) and intuitively, we found that the symbolic operation (i.e., 1-bit quantization) helps alleviate the heterogeneity of model updates, since all updates then have a uniform magnitude across all the dimensions of the model for each client. Furthermore, reducing model heterogeneity should also intuitively contribute to improving the model performance in heterogeneous federated settings.
> > >
> > > However, such a sign operation (e.g., QSGD) alone does not directly improve the generalization in experiments. Inspired by the Lion optimizer [CC1]  that incorporates the sign operation and then enhances the convergence and generalization in centralized learning, we thus introduced Lion’s structure into federated learning, and verifies that this incorporation can indeed improve the global model’s generalization performance.
> > >
> > > In fact, the reason why sign operations improve the generalization of centralized learning only with the Lion optimizer is itself a challenging problem to analyze in depth. This unique optimization algorithm was derived through a program search with a total cost of 3K TPU V2 days. The original Lion manuscript provides a simple explanation, suggesting that the sign operation introduces noise into the updates, acting as a form of regularization that aids in the generalization. In our analysis, we investigated the sign operation within the federated learning scenario and found that, in this context, the sign operation helps mitigate the model heterogeneity caused by data heterogeneity. This, in turn, may improve the overall performance of the globally aggregated model at the server.
> > >
> > > Many thanks again for the time and effort that you have dedicated to reviewing our paper and providing these insightful comments. In Appendix B.3 of the revised manuscript, we have also incorporated the above discussion as part of our future work.
> > >
> > > [CC1] Chen, Xiangning, et al. ''Symbolic discovery of optimization algorithms.'' *Advances in neural information processing systems* 36 (2024).

---

### Official Review · Reviewer_8BEn · 2024-11-04

**Soundness:** 3
**Presentation:** 3
**Contribution:** 3
**Rating:** 8
**Confidence:** 4

**Summary:**

Authors:
- Develop a compression-based FL algorithm called FedSMU.
- FedSMU uses the sign operation to achieve a 1-bit compression and thus greatly saves the communication cost.
- We conduct a convergence analysis of FedSMU under the general non-convex settings and dependence is optimal for such assumption in terms of T.
- Authors conducted a series of experiments to demonstrate the superiority of FedSMU

**Strengths:**

- originality: Handling signed information is interesting for distributed training from a theoretical perspective.
- quality: The text is written pretty in quality form.
- significance: Results are interesting for theoretical advancements.

**Weaknesses:**

In this section, I provide constructive and actionable insights on how the work could improve. However, they are not so critical:

**Minor Comments:**

q1: **Line 60:** The statement "However, most of these algorithms will lose a certain amount of information due to compression, resulting in a decrease in accuracy" needs clarification. If the authors mean this is true for a specific number of rounds/epochs, then it is accurate. However, in general, variance-reduced techniques do not necessarily lead to a loss of accuracy. The convergence guarantees for these methods are comparable to those without compression, although they may require more rounds. The authors should consider discussing the convergence rates relevant to specific situations.

q2: **Line 64:** The phrase "mitigate the client variance" lacks clarity. It would be beneficial to specify what type of variance is being addressed—variance in model updates, loss functions, or another context.

q3: **Line 68:** The claim "Additionally, it remains unknown whether these optimization-based algorithms are compatible with the current compression techniques" is misleading. If there exists a theorem demonstrating compatibility, then it is established. Typically, these theorems analyze convergence behavior on the training set, and generalization aspects may not be straightforward. Thus, stating that compatibility "remains unknown" is inaccurate, as compression techniques are an integral part of training algorithms in Federated Learning.

q4: **Line 85:** The term "Lion optimizer" is not properly referenced in the Introduction section.

----

**Middle Comments:**

----

q5: **Line 300:** The statement "FedLion requires the additional transmission of the full-precision momentum terms, resulting in a significantly higher communication cost compared to our FedSMU that only necessitates a 1-bit communication for each dimension of the model updates" lacks clarity regarding the necessity of momentum terms on the server side of Algorithm 1.

q6: Can momentum terms be stored on the client side instead?

----
**Request for Discussion:**


q7: I suggest that authors elaborate a bit on why "sign" information 1-bit can be useful for practice or imagine scenarios when it can be useful for society.

The service information for communication includes plenty of headers, for instance:
- The Ethernet header has 14 bytes for the header.
- IPv4 has 20 bytes header.
- TCP has header 20 header.

Problem-1: 1 bit can not be transferred physically, only 1 byte (approximately).
Problem-2: Even with transferring 1 byte you will need to transfer 54 bytes of headers.

**Questions:**

**Major Comments:**

Q1: I can not find "Supplementary Material" and I can not find "Link for code". Without this, I can not reproduce experiments. Please provide it.

Q2: Line 10: The statement "While biased compression with a certain degree of information loss may cause a slight decrease in model performance" is incorrect. Refer to methods like MARINA, DIANA, and EF21 for comparison do not have any inherited limitations with using compression. Please clarify your statement.

Q3: Line 309: The three standard assumptions stated in the analysis are weak and not universally applicable to a range of non-convex objective functions. See references to EF21 and MARINA for examples of relaxed analyses.

- Assumption 4.3 does not allow for the minimization of y=x^2/2 over R.
- Assumption 4.2 does not accommodate data-point subsampling.

I am asking the authors to elaborate on the weakness of assumptions for practical purposes.

Q4: Theorem 4.4: The convergence is established in terms of the $L_1$ norm, which is always greater than $L_2$. While your rates in terms of $L_1$ norm square are inversely proportional to $1/T$ you have a dependency of $d^2$, which becomes problematic.
Please elaborate more on the dependency on dimension. Your method is not Zero-Order to have a dependency on dimension (which for instance Convex Optimization setting has with the Zero-Order optimization class of Algorithms in terms of convergence to full gradient square https://arxiv.org/pdf/1312.2139, p.7)

Q5: State-of-the-art Methods for Top-K Compression: To my knowledge, the state-of-the-art method for Top-K compression is Error Feedback 2021, not FedEF-TopK. For reference, please see:
- https://arxiv.org/pdf/2106.05203
- https://arxiv.org/abs/2110.03294
- https://arxiv.org/abs/2402.10774

If you can for camera-ready make a comparison with EF21. It would be great. I believe all these Q1-Q5 are addressable.

Thanks for your work.

---

> ### Author Response · Authors · 2024-11-22
> **Response to Comments 1 to 6**
>
> We would like to thank the reviewer for the comments. In the following, we have provided our detailed responses to each of these comments.
>
> >Comment 1: Line 60: The statement ``However, most of these algorithms will lose a certain amount of information due to compression, resulting in a decrease in accuracy'' needs clarification.
>
> **Response:**
>
> Thanks for this comment. We will clarify these points in the revised manuscript, as follows:
>
> However, the direct use of compression methods will lose a certain amount of information, resulting in the problem of decrease in the accuracy [B1] and slower convergence or even divergence [B2]. On the other hand, some methods like Error Feedback [B3] have designed to solve these problems.
>
> [B1] Yu, Enda, et al. "CP-SGD: Distributed stochastic gradient descent with compression and periodic compensation.'' In *Journal of Parallel and Distributed Computing* 169 (2022): 42-57.
>
> [B2] Beznosikov, Aleksandr, et al. "On biased compression for distributed learning.'' In *Journal of Machine Learning Research* 24.276 (2023): 1-50.
>
> [B3] Richtárik, Peter, Igor Sokolov, and Ilyas Fatkhullin. "EF21: A new, simpler, theoretically better, and practically faster error feedback.'' In *Advances in Neural Information Processing Systems* 34 (2021): 4384-4396.
>
> >Comment 2: Line 64: The phrase "mitigate the client variance'' lacks clarity.
>
> **Response:**
>
> Please note that the variance here refers to the variance between the uploaded client updates. Building on the SAGA algorithm in centralized learning, SCAFFOLD reduces the variance between the uploaded client updates (the sum of gradients) by employing control variates. As highlighted in the Abstract of the original SCAFFOLD manuscript: "As a solution, we propose a new algorithm (SCAFFOLD) which uses control variates (variance reduction) to correct for the ‘client-drift’ in its local updates." Furthermore, as mentioned in the Appendix of the original manuscript, "Our method can be viewed as seeking to remove the ‘client-variance’ in the gradients across the clients."
>
> >Comment 3: Line 68: The claim "Additionally, it remains unknown whether these optimization-based algorithms are compatible with the current compression techniques" is misleading.
>
> **Response:**
>
> Please note that integrating federated optimization techniques with federated compression algorithms poses significant challenges in achieving good convergence rates while simultaneously reducing communication overhead. For instance, SCALLION [B4] introduces variance-reduced optimization techniques to achieve a faster convergence and reduced upload communication. However, this comes at the cost of doubling the download communication overhead due to the transmission of control variables.
>
> [B4] Huang, Xinmeng, Ping Li, and Xiaoyun Li. "Stochastic controlled averaging for federated learning with communication compression." *arXiv preprint arXiv:2308.08165* (2023).
>
> >Comment 4: Line 85: The term "Lion optimizer'' is not properly referenced in the Introduction section.
>
> **Response:**
>
> Thanks for pointing out this out issue, and we will include the appropriate reference for the Lion optimizer in the revised manuscript.
>
> >Comment 5: Line 300: The statement  lacks clarity regarding the necessity of momentum terms on the server side of Algorithm 1.
>
> **Response:**
>
> We would like to clarify that our FedSMU (Algorithm 1) does not need to store the momentum terms on the server side, and only tracks the momentum state on the client side. As a result, FedSMU only requires uploading the 1-bit model update after symbolization for each client. In contrast, FedLion requires to aggregate and store momentum on the server side. Consequently, FedLion necessitates the additional transmission of the full-precision momentum terms, along with the model update after quantization.
>
> >Comment 6: Can momentum terms be stored on the client side instead?
>
> **Response:**
>
> Please note that in our FedSMU algorithm, each client needs to maintain and track its momentum state, but does not upload it to the server. While these momentum states may become somewhat stale due to partial participation, their impact on the performance is minimal and acceptable, as demonstrated in Table 3 of the original manuscript.

---

> ### Author Response · Authors · 2024-11-22
> **Response to Comment 7 and Questions 1 and 2**
>
> >Comments 7: I suggest that authors elaborate a bit on why ``sign'' information 1-bit can be useful for practice or imagine scenarios when it can be useful for society. 1 bit can not be transferred physically, only 1 byte (approximately). Even with transferring 1 byte you will need to transfer 54 bytes of headers.
>
> **Response:**
>
> We would like to clarify that the "sign operation'' refers to 1 bit per dimension, not a total of 1 bit for the entire model. In practice, the weights or gradients of each dimension in a model are typically stored as 32-bit floating-point values (float32). This means that each dimension requires 32 bits to upload. However, when applying the sign operation, each dimension only requires 1 bit to represent the sign, thus significantly reducing the communication overhead compared to using 32 bits per dimension.
>
> In terms of the physical data transfer, the actual number of bits transmitted to the server can be calculated as:
> $\text{Total bits transferred} = (\text{bits per dimension} \times \text{the number of dimensions} \times \text{the number of participating clients})$ + the header size. For example, using a CNN model with $n = 10$ clients, FedSMU would transfer approximately 0.95MB. Given the large volume of transmitted data, the header size of 54 bytes is negligible in comparison.
>
> >Question 1: I can not find "Supplementary Material'' and I can not find "Link for code".
>
> **Response:**
>
> We provided an Appendix to the main text and thus have no additional supplementary material, and the code for evaluating our FedSMU has been provided in the following anonymous link <https://anonymous.4open.science/r/fedsmu-400D>.
>
> >Question 2: Line 102: The statement ``While biased compression with a certain degree of information loss may cause a slight decrease in model performance'' is incorrect.
>
> **Response:**
>
> Thanks for this comment, and we will make the following clarification in the revised manuscript.
>
> A direct application of biased compression on gradient into federated learning, such as Top-k, can lead to a performance degradation and lower convergence speed due to bias accumulation [B2]. To address this, some studies have introduced optimization techniques to mitigate the negative effects of bias. For example, EF21 [B3] employs the error feedback, while MARINA[B5] and DIANA[B6] leverage the compression of gradient differences, both of which further enhance the model performance and convergence speed.
>
> [B5] Gorbunov, Eduard, et al. "MARINA: Faster non-convex distributed learning with compression.'' In *International Conference on Machine Learning*. PMLR, 2021.
>
> [B6] Mishchenko, Konstantin, et al. "Distributed learning with compressed gradient differences.'' In *Optimization Methods and Software* (2024): 1-16.

---

> ### Author Response · Authors · 2024-11-22
> **Response to Questions 3 to 5**
>
> >Question 3: Line 309: The three standard assumptions stated in the analysis are weak and not universally applicable to a range of non-convex objective functions.
>
> **Response:**
>
> Please note that Assumption 4.1 is a standard assumption in the convergence analysis of optimization algorithms. Similar assumptions can be found in MARINA ("Assumption 1.1'') and EF21 ("Assumption 1'').
>
> For Assumption 4.2, it is fundamental for SGD-based optimization algorithms. In SGD, model updates are computed using mini-batch sampling rather than full-batch, under the assumption that the sampling process is unbiased and accounts for the variance introduced. However, MARINA bypasses this assumption by employing the Gradient Descent (GD) instead of Stochastic Gradient Descent (SGD), eliminating the variance caused by mini-batch sampling. EF21 discuss this stochastic gradient explicitly in Section ``F: Dealing with Stochastic Gradients (Details for Section 3.6)''.
>
> For Assumption 4.3, we acknowledge that this is a stronger assumption, ensuring that both the compressed targets and the momentum term (the moving averages of gradients) in our theoretical analysis are bounded. Moreover, this assumption has been adopted in other federated optimization and sign-based compression studies [B7,B8]. Specifically, in [B7], the authors demonstrate the convergence of distributed SIGNSGD with momentum under Assumption 4, which is also the bounded gradient assumption.
>
> Regarding the author's mention of $y = \frac{x^2}{2}$ over $\mathbb{R}$, it is true that the gradient does not have an upper bound for the domain $\mathbb{R}$. However, in neural networks, the input domain is typically bounded, ensuring the gradient is also bounded. For instance, gradient clipping is a commonly used technique to control the gradient's upper bound and prevent gradient explosion.
>
> We will clarify these points in the revised manuscript.
>
> [B7] Tao Sun, Qingsong Wang, Dongsheng Li, and Bao Wang. "Momentum ensures convergence of signsgd under weaker assumptions.'' In *International Conference on Machine Learning*, pp. 33077–33099. PMLR, 2023.
>
> [B8] Sashank Reddi, Zachary Charles, Manzil Zaheer, Zachary Garrett, Keith Rush, Jakub Konecny, Sanjiv Kumar, and H Brendan McMahan. "Adaptive federated optimization.'' *arXiv preprint arXiv:2003.00295*, 2020.
>
> >Question 4: Theorem 4.4: While your rates in terms of $L_1$ norm square are inversely proportional to $\frac{1}{T}$ you have a dependency of $d^2$ , which becomes problematic. Please elaborate more on the dependency on dimension.
>
> **Response:**
>
> **The convergence rate in terms of $L_1$ norm with a dependency of $d$ is common in sign-based method, like Theorem 4 in literature[B7], p.6, Theorem 1 in literature [B9], P.6, and Equation 35 in literature [B10], P.19.** Their form of expression is similar to Theory 4.4. However, it is worth noting that the value of the learning rate $\gamma_1$ may be critical. In literature [B9] and [B10], the authors use $\gamma_1=\frac{1}{\sqrt{Td}}$ to reduce the power of $d$. In that way, if we take $\gamma_1=\frac{1}{L\sqrt{Td}}$, Theorem 4.4 can only depend on $\sqrt{d}$.
>
> [B9] Chen, Xiangyi, et al. "Distributed training with heterogeneous data: Bridging median-and mean-based algorithms." In *Advances in Neural Information Processing Systems* 33 (2020): 21616-21626.
>
> [B10] Jin, Richeng, et al. "Stochastic-sign SGD for federated learning with theoretical guarantees." *arXiv preprint arXiv:2002.10940* (2020).
>
> >Question 5: To my knowledge, the state-of-the-art method for Top-K compression is Error Feedback 2021, not FedEF-TopK.
>
> **Response:**
>
> Thanks for this suggestion, and we have additionally conducted a comparison with the original EF21 algorithm. Due to the limited time, only experiments on CIFAR-10 and CIFAR-100 are conducted. We set a total of 100 clients with different participation rate (0.03\% and 0.1\%, represented in Table B1 by L and H) and use Dirichlet-0.25. The experimental results are shown in Table B1.
>
> On CIFAR-100, FedSMU still shows a high performance. While on CIFAR-10, the accuracy of FedSMU can be higher than EF21 with a lower participation. These results strongly demonstrate the superiority of FedSMU in complex image classification tasks, especially under a low client participation rate, which may result from the sign operation promoting the fair contribution of clients effectively to the global model update.
>
> **Table B1: Performance comparison under different datasets. L and H indicate low and high participation rates. Bold numbers indicate the best performance.**
>
> | **Dataset**   | **Setting**       | **FedSMU** | **EF21**  |
> |---------------|-------------------|------------|-----------|
> | **CIFAR-10**  | Dir(0.25)-L       | **79.66**  | 74.43     |
> |  **CIFAR-10**   | Dir(0.25)-H       | 80.67      | **81.51** |
> | **CIFAR-100** | Dir(0.25)-H       | **51.79**  | 50.07     |

---

> ### Comment · Reviewer_8BEn · 2024-11-26
> **Thanks**
>
> I appreciate the authors' thorough rebuttal and commend them for addressing the majority of my concerns. I encourage them to revise the paper accordingly and integrate the new discussions into the camera-ready version.  All your comments I believe is important to include in the extra discussion section in the Appendix for clarity. Thanks for your hard work.

---

> > ### Author Response · Authors · 2024-11-28
> > **Official Comment by Authors**
> >
> > Dear Reviewer 8BEn:
> >
> > We sincerely appreciate your time and effort in reviewing our paper and providing the valuable comments. We have incorporated these additional discussions and new experiments into the revised manuscript that was newly uploaded. Specifically, new baselines, including EF21, have been included in the Appendix G.4.
> >
> > For the summary of our changes in the revised manuscript, please refer to our General Response. We believed that these insights will further contribute to enhancing the quality of the final version of our manuscript.

---

### Official Review · Reviewer_8rqU · 2024-11-04

**Soundness:** 2
**Presentation:** 3
**Contribution:** 2
**Rating:** 5
**Confidence:** 3

**Summary:**

The paper introduces FedSMU, a federated variant of the Lion optimizer, designed to address communication efficiency in federated learning (FL). The FedSMU algorithm uses the Lion optimizer's sign-based updates, reducing client-to-server communication costs to 1-bit per parameter. The paper also provides a convergence analysis for both FedSMU and the original Lion optimizer, and compares to other compression-based methods.

**Strengths:**

1- Compared to FedLion, the design of FedSMU is better suited for FL, specifically in handling communication constraints. FedSMU has a communication cost of only 1-bit per parameter from the client to the server. On the other hand, FedLion incurs even higher communication costs than FedAvg and shows weaker performance in practice compared to FedSMU.

2- By sending only 1-bit per parameter, FedSMU balances the magnitude of updates between different clients, which is beneficial for handling data heterogeneity. The paper also provides strong motivation for this choice.

3- In experimental evaluations, FedSMU performs better than other communication-efficient FL methods, especially in settings with high data heterogeneity.

**Weaknesses:**

Major: Missing related work: FedSMU is really similar to the Distributed Lion Training [1]. Technically, it's the same method with standard changes (Local steps and Partial client participation). This makes the novelty of the paper limited. Additionally, this paper (in addition to averaging) also considers majority vote for aggregating the client updates, and it seems that it works better in practice.

[1] Liu, Bo, et al. "Communication Efficient Distributed Training with Distributed Lion." arXiv preprint arXiv:2404.00438 (2024).

----------------------
1- The paper acknowledges that clients in FL can dynamically join or leave. However, the FedSMU algorithm requires clients to maintain and track their momentum state, which may introduce practical limitations in scenarios with dynamic participation.

2- Given the comparison with SCAFFOLD on CIFAR-10 and Shakespeare datasets, benchmarking on additional datasets and model architectures could strengthen the paper’s results.

3- Additional experiments comparing methods based on both communication cost and the number of rounds would provide clearer insights into practical efficiency. Comparisons with SCAFFOLD, FedAvg, etc., based on communication cost would be helpful. Comparison in terms of the number of rounds  vs compression-based method and non-compression-based methods also missing. Communication cost is an important factor, however the number of rounds in which clients need to compute gradients is also important.

4- Given the paper’s claims regarding generalization improvement, experiments that explicitly measure generalization (e.g., by comparing validation performance at similar training error levels) are needed to substantiate this claim further.

**Questions:**

1- Can you plot the accuracy vs MU for different methods, checkpoints and dataset to better show the correlation between them?

2- How does your model compare to other methods in terms of wall-clock time? (A balance for comparing number of rounds needed vs communication costs.)

3- Can you include compression for server-to-client communication as well? A comparison to some baselines can be helpful.

4- Can the convergence bound improve with more number of local steps and more participation per round?

5- Can this method improve over SCAFFOLD in CIFAR-10 and Shakespeare with more communication (more rounds or other variant like FedSMUMC)?

---

> ### Author Response · Authors · 2024-11-22
> **Response to Major Comment and Comment 1**
>
> We would like to thank the reviewer for the comments. In the following, we have provided our detailed responses to each of these comments.
>
> >Major Comment:  Missing related work: FedSMU is really similar to the Distributed Lion Training.
>
> **Response:**
>
> Thank you for this comment. We will refer to the "Distributed Lion Training'' paper as [A1] in this response and the revised manuscript, and clarify the differences and advantages of our FedSMU compared to the D-Lion in [A1], as follows.
>
> **1) Motivation.** Our FedSMU can simultaneously mitigate data heterogeneity and reduce communication compression through the symbolic operations. The analysis was carried out and verified by experiments (Figure 1). While D-Lion only consider to compress the communication.
>
> **2) Scope of application.** Our FedSMU can deal with scenarios involving the partial client participation and multiple local updates, whereas D-Lion can not. Performing multiple local updates, in the federated settings, can effectively reduce the communication frequency and thus the overall traffic. Experimental results in Table A1 further demonstrate that D-Lion fails in such scenarios with low client participation rates and multiple local updates, whereas FedSMU remains robust and performs well under these conditions.
>
> **Table A1: Performance comparison under different datasets. F and P indicate full and partial participation rates, and K is the number of local updates. Bold numbers indicate the best performance.**
>
> | **Algorithm** | **K=1 with F** | **K=5 with F** | **K=5 with P** |
> |---------------|----------------|----------------|----------------|
> | FedSMU        | **81.61**      | **80.54**      | **82.08**      |
> | D-Lion        | 68.83          | 55.19          | 54.34          |
>
> **3) Algorithms design.** While both algorithms are based on the Lion optimizer, FedSMU fully leverages the structural advantages of the Lion optimizer, including weight decay in the global aggregation. In contrast, D-Lion primarily incorporates the momentum sliding averaging and symbolic operations at local update. Notably, when the number of local updates and clients is set to 1, our algorithm reduces to the standard Lion optimizer, whereas D-Lion does not. This comprehensive utilization of the Lion optimizer structure may explain why the experimental performance of our FedSMU surpasses that of D-Lion.
>
> **4) Compatibility with majority vote.** We have further extended FedSMU with majority vote, as FedSMU-MV. Experimental results in Table A2 show that FedSMU-MV achieves an accuracy of 47.66\% on CIFAR-100, slightly lower than FedSMU's 51.79\% under the same settings (number of clients = 100, fraction = 0.1, Dirichlet = 0.25). This indicates that majority vote is compatible with our algorithm. The slight accuracy drop may result from FedSMU's symbolic model updates. Applying majority vote to the 1-bit results could further suppress some clients' model update information due to the dominant update direction.
>
> **Table A2: Performance comparison on CIFAR-100. Bold numbers indicate the best performance.**
>
> | **Dataset**   | **FedSMU** | **FedSMU-MV** |
> |---------------|------------|---------------|
> | CIFAR-100     | **51.79**  | 47.66         |
>
> [A1] Liu, Bo, et al. "Communication Efficient Distributed Training with Distributed Lion.'' *arXiv preprint arXiv:2404.00438* (2024).
>
> >Comment 1:  FedSMU algorithm requires clients to maintain and track their momentum state, which may introduce practical limitations in scenarios with dynamic participation.
>
> **Response:**
>
> Our FedSMU algorithm requires each client to maintain and track its momentum state. In some extreme cases, where each client participates in the training only once during the entire training process, our algorithm may fail. And we will include this limitation in the revised manuscript.
>
> In fact, in the original manuscript, we did analyze this limitation from two perspectives. First, we tested our algorithm at a lower client participation rate (i.e., where the momentum state of the clients is more stale). The results in Table 4 show that FedSMU still achieves a good accuracy in most cases. Additionally, we introduce a variant algorithm, FedSMUC, which additionally store and communicate the complete momentum at the server at the cost of communication overhead. The results in Table 6 demonstrate that appropriately completing the momentum can marginally improve the model performance. Overall, the experimental results indicate that, in most cases, our FedSMU algorithm remains effective and does not fail due to the stale momentum.

---

> > ### Comment · Reviewer_8rqU · 2024-11-25
> > **Comparison with D-Lion**
> >
> > Thanks for your responses.
> >
> > I'm confused about the comparison with D-Lion. FedSMU with  k=1 and full participation is the same algorithm as D-Lion, and similarly, D-Lion also reduces to Lion when n=1. Why is there a huge performance gap in the table. Can you explain the details of the experiments and methods?

---

> > > ### Author Response · Authors · 2024-12-03
> > > **Continued Response to Comparison with D-Lion**
> > >
> > > Dear Reviewer 8rqU,
> > >
> > > We have completed the additional experiments on the CIFAR-100 dataset, with results shown in Table AA3 and Table AA4. These results remain consistent with our previous findings on the CIFAR-10 dataset, and we summarize the observations as follows.
> > >
> > > - **Performance under full participation or partial participation:** In scenarios with the full client participation, FedSMU demonstrates a slightly inferior performance compared to D-Lion. However, under the partial client participation, FedSMU consistently outperforms D-Lion.
> > >
> > > - **Impact of increasing number of local updates $(K)$:** With the heterogeneous data settings and full client participation, the performance of FedSMU surpasses that of D-Lion as the number of local updates increases.
> > >
> > > - **Comparison with FedAvg under low local updates $(K=1)$:** When the data is heterogeneous and the number of local updates is limited to one, both FedSMU and D-Lion exhibit a worse performance compared to FedAvg.
> > >
> > > We will incorporate these additional experimental results on CIFAR-100 into the final version of our submission.
> > >
> > > **Table AA3: Performance comparison on CIFAR-100 dataset, where F and P indicate full and partial participation, and $K$ is the number of local updates.**
> > > | **Setting**      | **Algorithm** | **K=1 with F** | **K=1 with P** | **K=5 with F** | **K=5 with P** |
> > > |-------------------|---------------|----------------|----------------|----------------|----------------|
> > > | **Dir-0.25**      | FedSMU        | 14.85          | 20.41          | **45.69**      | **42.06**      |
> > > |  **Dir-0.25**   | D-Lion        | 15.42          | 3.9            | 45.54          | 8.03           |
> > > |  **Dir-0.25**   | FedAvg        | **44.99**      | **39.34**      | 36.55          | 36.51          |
> > > | **iid**           | FedSMU        | 50.98          | **47.18**      | 49.76          | **49.72**      |
> > > | **iid**     | D-Lion        | **51.46**      | 5.13           | **50.11**      | 13.07          |
> > > | **iid**     | FedAvg        | 44.85          | 41.07          | 41.38          | 38.25          |
> > >
> > > **Table AA4: Performance comparison on CIFAR-100 dataset, where F and P indicate full and partial participation, and $K$ is the number of local updates.**
> > > | **Setting**      | **Algorithm** | **K=100 with F** | **K=100 with P** | **K=500 with F** | **K=500 with P** |
> > > |-------------------|---------------|----------------|----------------|----------------|----------------|
> > > | **Dir-0.25**      | FedSMU        | **50.15**          | **50.62**          | **46.66**      | **48.2**      |
> > > |  **Dir-0.25**   | D-Lion        | 49.84          | 4.05            | 46.55          |16.21          |

---

> ### Author Response · Authors · 2024-11-22
> **Response to Comments 2 and 3**
>
> >Comment 2: Given the comparison with SCAFFOLD on CIFAR-10 and Shakespeare datasets, benchmarking on additional datasets and model architectures could strengthen the paper’s results.
>
> **Response:**
>
> Thanks for your suggestion, and we have compared our FedSMU with SCAFFOLD on new MNIST dataset. Additionaly, we also conducted another performance comparison scenario ($Dir = 0.6$) on CIFAR-100 using ResNet-18. The experimental results in Table A3 show that FedSMU outperforms SCAFFOLD on CIFAR-100. However, for the simpler MNIST dataset, FedSMU performs slightly worse than SCAFFOLD. This slight degradation on MNIST is likely due to the lower degree of heterogeneity in the dataset.
>
> In a federated heterogeneous scenario with CIFAR-100, which consists of 100 categories compared to MNIST's 10 categories, each client typically handles a subset of 13-16 or 20-25 categories when \(Dir = 0.25\) or \(Dir = 0.6\), respectively. This high degree of heterogeneity in CIFAR-100 leads to greater deviations in model updates among clients, enabling FedSMU to achieve more significant improvements compared to MNIST. To further assess performance, we plan to explore a more complex dataset, Tiny ImageNet, for additional comparisons in future work.
>
> **Table A3: Performance comparison under different datasets. L and H indicate low and high participation rates. Bold numbers indicate the best performance.**
>
> | **Dataset (Model)**        | **Setting**       | **SCAFFOLD** | **FedSMU**  |
> |----------------------------|-------------------|--------------|-------------|
> | **MNIST on LeNet**         | Dir (0.6)-L       | **98.4**     | 97.37       |
> |   **MNIST on LeNet**           | Dir (0.6)-H       | **98.2**     | 97.47       |
> | **CIFAR-100 on ResNet18**   | Dir (0.6)-H       | 53.90        | **54.25**   |
>
> >Comment 3: Additional experiments comparing methods based on both communication cost and the number of rounds would provide clearer insights into practical efficiency.
>
> **Response:**
>
> Thank you for the suggestion. We will include other algorithms in the figures depicting convergence performance vs. the number of communication bits. Additionally, we will provide new figures depicting convergence performance vs. the number of communication rounds in the Appendix.
>
> Here since we cannot show the figures directly, we compare the convergence rate using the number of communication rounds required to achieve the target accuracy. The results are presented in the Table A4. For CIFAR-100 and Shakespeare, our algorithm does not require more communication rounds compared to most algorithms. However, for CIFAR-10, it slightly exceeds the number of rounds needed by other algorithms. This may be attributed to the simplicity of the CIFAR-10 dataset, which has a lower degree of heterogeneity. In this case, our algorithm's strengths are not fully utilized, as the initial training process is already close to convergence.
>
> **Table A4: Number of communication rounds to achieve a preset target accuracy. "/" means it cannot reach the training accuracy.**
>
> | **Dataset**              | **Training Accuracy (%)** | **FedAvg** | **SCAFFOLD** | **SCALLION** | **FedEF-HS** | **FedEF-TopK** | **FedEF-Sign** | **FedLion** | **FedSMU**  |
> |--------------------------|---------------------------|------------|--------------|--------------|--------------|----------------|----------------|-------------|-------------|
> | **CIFAR-10**             | 55                        | 26         | 29           | 23           | 41           | 37             | 50             | **21**      | 65          |
> |  **CIFAR-10**            | 60                        | 43         | 43           | **29**       | 62           | 57             | 81             | 33          | 118         |
> |  **CIFAR-10**        | 65                        | 57         | 62           | 48           | 108          | 96             | 111            | **50**      | 288         |
> | **CIFAR-100**            | 35                        | 193        | **86**       | 286          | 690          | 355            | 794            | 100         | 832         |
> |  **CIFAR-100**   | 40                        | 629        | **142**      | 730          | /            | 882            | /              | 225         | 1218        |
> |   **CIFAR-100**   | 45                        | /          | **270**      | 3703         | /            | /              | /              | 632         | 1811        |
> | **Shakespeare**          | 25                        | 17         | 12           | 13           | 30           | 20             | 57             | **10**      | 11          |
> |   **Shakespeare**   | 30                        | 32         | 19           | 20           | 45           | 36             | 78             | **17**      | 20          |
> |    **Shakespeare**    | 35                        | 61         | **27**       | **27**       | 85           | 68             | 177            | 28          | 48          |

---

> ### Author Response · Authors · 2024-11-22
> **Response to Comment 4 and Questions 1 and 2**
>
> >Comment 4: Experiments that explicitly measure generalization (e.g., by comparing validation performance at similar training error levels) are needed to substantiate this claim further.
>
> **Response:**
>
> Thank you for this comment. We will further evaluate the performance of our algorithm in terms of generalization mentioned. Based on this definition, we compare validation performance at similar training accuracy levels.
>
> Due to time limitations, we conduct comparisons under the following settings: CIFAR-10 and CIFAR-100 datasets: 100 users, 0.1 participation rate, and Dirichlet = 0.25.
>
> The experimental results are shown in the Table A5. The results show that on the CIFAR-10 and CIFAR-100 datasets, FedSMU achieves the highest test accuracy and demonstrates the best generalization performance.
>
> **Table A5: Generalization performance comparison under various datasets. "/" means it cannot reach the training accuracy. Bold numbers indicate the best performance.**
>
> | **Dataset**        | **Training Accuracy** | **FedAvg** | **SCAFFOLD** | **SCALLION** | **FedEF-HS** | **FedEF-TopK** | **FedEF-Sign** | **FedLion** | **FedSMU**  |
> |--------------------------|-----------------------|------------|--------------|--------------|--------------|----------------|----------------|-------------|-------------|
> | **CIFAR-10**             | 83-84                 | 75.14      | **77.49**    | 75.9         | 75.63        | 75.48          | 75.7           | 77.22       | 77.1        |
> | **CIFAR-10**      | 85-86                 | 76.17      | 76.37        | 76.87        | 76.83        | 76.83          | 77.4           | 78.3        | **78.39**   |
> |  **CIFAR-10**   | 87-88                 | 78.56      | 79.24        | 77.79        | 77.85        | 77.74          | /              | 79.23       | **79.78**   |
> | **CIFAR-100**            | 66-67                 | 40.08      | 45.76        | 40.56        | /            | 41.28          | /              | 45.55       | **49.03**   |
> |  **CIFAR-100**    | 68-69                 | 40.6       | 46.16        | 40.88        | /            | 41.81          | /              | 46.15       | **50.04**   |
> |  **CIFAR-100**     | 70-71                 | 40.9       | 46.76        | 41.23        | /            | 42.34          | /              | 46.56       | **50.85**   |
>
> >Question 1: Can you plot the accuracy vs MU for different methods, checkpoints and dataset to better show the correlation between them?
>
> **Response:**
>
> Thank you for the suggestions. We will include the plots in the revised manuscript. Meanwhile, here, we provide Table A6 and Table A7 showing the correlation between Magnitude Uniformity (MU), data heterogeneity, and accuracy. The results indicate that with FedAvg, data heterogeneity significantly amplifies the differences in the magnitude of model updates across clients, leading to unstable global aggregation and poorer generalization performance. While SCAFFOLD reduces variance to address these differences, FedSMU directly ensures consistency across all model updates through symbolic operations. Those two approaches enhance Magnitude Uniformity among clients, ultimately improving accuracy.
>
> **Table A6: Accuracy (%) vs. data heterogeneity.**
>
> | **Algorithm**  | **Dirichlet-0.25** | **Dirichlet-0.6** | **Dirichlet-0.8** | **iid**   |
> |----------------|--------------------|-------------------|-------------------|-----------|
> | **FedSMU**     | 51.79              | 53.49             | 54.21             | 55.92     |
> | **FedAvg**     | 41.44              | 43.44             | 44.21             | 48.05     |
> | **SCAFFOLD**   | 50.49              | 50.02             | 53.89             | 54.78     |
>
> **Table A7: MU vs. data heterogeneity**
>
> | **Algorithm**  | **Dirichlet-0.25** | **Dirichlet-0.6** | **Dirichlet-0.8** | **iid**   |
> |----------------|--------------------|-------------------|-------------------|-----------|
> | **FedSMU**     | 192.00             | 192.00            | 192.00            | 192.00    |
> | **FedAvg**     | 46.98              | 48.00             | 48.32             | 55.46     |
> | **SCAFFOLD**   | 84.25              | 87.41             | 90.15             | 90.86     |
>
> >Question 2: How does your model compare to other methods in terms of wall-clock time?
>
> **Response:**
>
> We have tested the wall-clock time needed for each baseline to execute one communication round. Take CIFAR-100 and participation rate $\frac{n}{m}=0.1$ as an example, the average wall-clock time required to execute a round is as follows: FedSMU (10.43 seconds), FedAvg (10.15 seconds), FedEF-HS (10.46 seconds), FedLion (10.63 seconds), SCAFFOLD (10.38 seconds). Experiments demonstrate that in a single communication round, our algorithm introduces no significantly additional time overhead compared to other algorithms. As highlighted in the response to Comment 3, FedSMU does not require more communication rounds, ensuring no increase in overall training time compared to other algorithms.

---

> ### Author Response · Authors · 2024-11-22
> **Response to Questions 3 to 5**
>
> >Question 3: Can you include compression for server-to-client communication as well? A comparison to some baselines can be helpful.
>
> **Response:**
>
> Thanks for this comment. Due to the partial participation characteristic of federated learning, the server must broadcast the new global model, rather than a simple global model update, to initialize newly participating clients. This limitation prevents the direct application of uploaded model update compression techniques in our FedSMU to the downloaded global model.
>
> Following this advice, we will consider some model lightweight techniques, such as mixed-precision model compression, as a promising future research direction to compress the server-to-client communication in our FedSMU algorithm.
>
> >Question 4: Can the convergence bound improve with more number of local steps and more participation per round?
>
> **Response:**
>
> In the theoretical analysis of FedSMU, $\tau_{max}$ and $K$ control the client participation rate and the number of local updates, respectively. The convergence rate in Theorem 4.4 demonstrates that increasing the client participation rate and reducing the number of local updates can tighten the convergence bound.
>
> It is intuitive that a increased participation rate can improve the model's convergence rate. A higher client participation ratio helps the global model gather more information from the clients, which reduces the overfitting and mitigates the impact of local data heterogeneity, thereby improving the convergence bound. This conclusion is also supported by Table 1 in reference [A2].
>
> On the other hand, in the convergence upper bound of Theorem 4.4, the term $4K\sigma_l^2 + 4KG^2$ indicates that more local updates amplify both the local sampling variance $\sigma_l$ and the gradients accumulation $G$, resulting in a slower convergence. This aligns with our intuition. For the SGD-based local updates, as the number of local updates increases, the cumulative sampling variance $\sigma_l$ and gradients $G$ grows, also further leading to more divergent update directions caused by data heterogeneity. This negative effect is also experimentally demonstrated in Page 7 of literature [A2], where Figure 3 strongly states that ``for very large numbers of local epochs, FedAvg can plateau or diverge.''
>
> [A2] McMahan, Brendan, et al. "Communication-efficient learning of deep networks from decentralized data.'' In *Artificial intelligence and statistics*. PMLR, 2017.
>
> >Question 5: Can this method improve over SCAFFOLD in CIFAR-10 and Shakespeare with more communication (more rounds or other variant like FedSMUMC)?
>
> **Response:**
>
> We sincerely appreciate the reviewer for this suggestion. On CIFAR-10 dataset, the performance of FedSMU can slightly improve with more communication rounds, while SCAFFOLD's performance has been converged already.
>
> Here, specifically, we reconduct the experiment by increasing the number of rounds to 6000 and comparing FedSMUCS with SCAFFOLD on CIFAR-10 dataset, using a total of 100 clients with a partial participation ratio of 0.03 (represented in the Table A8 by L).
>
> The experimental results are shown in the Table A8. It can be found that on the CIFAR-10 dataset with Dir (0.6)-L, FedSMU achieves a slightly higher accuracy compared SCAFFOLD when the number of communication rounds reaches 6000. We also find that when FedSMUMC is used at additional overhead of storing and communicating the momentum, it can perform better than SCAFFOLD over the same 4000 rounds.
>
> However, it shoud be noted that there is a tradeoff between the training time and test accuracy. A larger number of communication rounds leads to a slight performance improvement. For example, with FedSMU on CIFAR-10 with Dir (0.6)-L, an accuracy of 81.96\% is achieved after 4000 rounds in the original manuscript, and an improvement of 0.52\% is obtained with 2000 more rounds (6000 rounds in total), which exceeds that of SCAFFOLD. Therefore, we currently set 4000 rounds as the maximum communication round for all the algorithms considering the training efficiency. We will incorporate this discussion into the revised manuscript.
>
> **Table A8: Performance comparison under different datasets**
>
> | **Dataset**  | **Setting**       | **Round** | **SCAFFOLD** | **FedSMU** | **FedSMUMC** |
> |--------------|-------------------|-----------|--------------|------------|--------------|
> | **CIFAR-10** | Dir(0.25)-L    | 4000     | **81.6**     | 79.66      | 80.47        |
> |**CIFAR-10**   | Dir(0.25)-L | 6000      | **81.6**     | 80.12      | 81.1         |
> | **CIFAR-10**  | Dir(0.6)-L     | 4000      | 82.36        | 81.96      | **82.67**    |
> |  **CIFAR-10**   |  Dir(0.6)-L     | 6000      | 82.36        | **82.48**  | **83.23**    |

---

> ### Author Response · Authors · 2024-11-28
> **Response to Comparison with D-Lion**
>
> Dear Reviewer 8rqU,
>
> Thanks for your reply and concern on the further comparison with D-Lion.
>
> Here, we would like to first make a correction to an ambiguity term that we might have incurred in our initial ``Response to Major Comment''. Specifically, instead of representing the expected meaning of number of local updates, the notation $K$ in Table A1 refers actually to the number of epoch updates in our experimental setup, where one epoch corresponds to a complete pass through the dataset. For example, if a client has 5000 samples and the minibatch size is set to 50, then it requires 100 updates within one epoch, and thus the actual $K$ in Table A1 should in fact correspond to 100 local updates. In this response, we have corrected this ambiguity term, and re-implemented the experiments accordingly to show a detailed comparison between our FedSMU and D-Lion.
>
> On the other hand, in our initial response, we only took the default hyperparameter settings from the original manuscript [A1] for evaluating D-Lion due to the time constraint. In this response, we have further conducted an extensive grid search and fully tuned the hyperparameters of D-Lion. Below, we provide the details of the hyperparameters used in our experiments.
>
> - To ensure a fair comparison, both algorithms are evaluated on the ClFAR-10 dataset  (we will also provide the CIFAR-100 results before the response deadline)`, using non-llD and IID data, with a total of 10 clients anda batch size of 50.
>
> - For FedSMU and FedAvg, we adopt the same parameter settings as outlined in Appendix A.
>
> - For D-Lion, we performed a grid search. The learning rate ($\epsilon$) is selected from {0.00005, 0.0005, 0.005, 0.015}, the weight decay ($\lambda$) is chosen from {0.0005, 0.005, 0.001, 0.01}, and $\beta_1$, $\beta_2$ are selected from {0.9, 0.99}. For Table AA1, the selected values are $\epsilon=0.015$, $\lambda=0.01$, $\beta_1=0.9$, $\beta_2=0.9$. For Table AA2, the selected values are $\epsilon=0.0005$, $\lambda=0.001$, $\beta_1=0.9$, $\beta_2=0.99$.
>
> **Table AA1 (Corrections to Table A1 in our initial response): Performance comparison on CIFAR-10 dataset, where F and P indicate full and partial participation rates, and K is the number of local updates.**
>
> | Setting     | Algorithm | $K=100$ ($K'=1$ epoch) with F | $K=500$ ($K'=5$ epochs) with F | $K=500$ ($K'=5$ epochs) with P |
> |-------------|-----------|------------------------------|-------------------------------|-------------------------------|
> | **Dir-0.25** | FedSMU    | **82.24**                    | **82.0**                      | **82.08**                     |
> | **Dir-0.25**            | D-Lion    | 82.19                        | 81.6                          | 51.23                         |
>
> ---
>
> **Table AA2: Performance comparison on CIFAR-10 dataset, where F and P indicate full and partial participation rates, and K is the number of local updates.**
>
> | Setting    | Algorithm | $K=1$ with F | $K=1$ with P | $K=5$ with F | $K=5$ with P |
> |------------|-----------|--------------|--------------|--------------|--------------|
> | **Dir-0.25** | FedSMU    | 32.47        | 38.35    | **77.97**    | **75.14**    |
> |**Dir-0.25** | D-Lion    | 34.03        | 24.58        | 77.62        | 34.48        |
> | **Dir-0.25** | FedAvg    | **79.64**    | **74.99**           |72.02           | 71.48          |
> | **iid**     | FedSMU    | 81.84        | **77.99**    | **82.37**    | **81.71**    |
> | **iid**   | D-Lion    | **82**       | 29.06        | 82.36        | 44.05        |
> |  **iid**     | FedAvg    | 79.53        | 76.7            | 76.3            | 75.84           |

---

> ### Author Response · Authors · 2024-11-28
> **Continued Response to Comparison with D-Lion**
>
> With these two modifications, we then present the additional experimental results in Table AA1 and Table AA2, with the following observations. Notably, since we have also fully tuned our FedSMU algorithm, we observe a slight improvement in its performance compared to the results in Table AA1 in our initial response.
>
> - With full participation and one local update (i.e., $K=1$ with F), FedSMU performs slightly worse than D-Lion. However, in scenarios with a partial participation, FedSMU consistently outperforms D-Lion. This is intuitive, as D-Lion does not maintain a complete global model at the server and only aggregates the global model updates. Thus in the partial participation settings, asynchronous clients can only save a stale global model. As a result, these clients may receive the global model updates, which, however, cannot be leveraged to recover the exact global model of the current round.
>
> - With multiple local updates (i.e., $K>1$), FedSMU consistently outperforms D-Lion. This performance improvement can be attributed to the different approaches to weight decay. Specifically, the hyperparameter $\gamma_2$ (denoted as $\lambda$ in D-Lion) controls the weight decay (or $L_2$ penalty) coefficient. In FedSMU, the regularization is applied to the global model $x_t$, potentially mitigating overfitting and thus enhancing generalization. In contrast, D-Lion applies this regularization to the local model $x_{t-1}^i$. As a result, when the local updates occur multiple times, D-Lion's regularization primarily affects the local model, and does not directly improve the generalization capability of the global model. Consequently, when finally evaluating the generalization performance of the global model, FedSMU demonstrates a significant advantage over D-Lion.
>
> - In heterogeneous scenarios, the performance of both FedSMU and D-Lion is poorer than that of FedAvg, especially when $k$ is small. This is an interesting and somewhat unexpected finding, which we speculate is due to the data heterogeneity. In the heterogeneous settings, each client samples a mini-batch of data for training and performs only a single time of update, followed by the application of the sign operation to the model update. Since the local update occurs only once, it introduces a substantial sampling variance and inter-client variance. The sign operation, which normalizes the magnitude of updates, may inadvertently amplify this variance between clients, leading to an unstable or even divergent global model aggregation.
>
> Finally, we would like to extend our heartfelt gratitude once again to the reviewer for the time, effort, and meticulous attention to detail in reviewing our paper. We kindly invite the reviewer to read our response, and look forward to further possible communication on the remaining issues.

---

### Author Response · Authors · 2024-11-28
**General Response**

First of all, we would like to express our gratitude to all the reviewers for providing the comments. Many of these valuable suggestions have helped us strengthen the rigor of our arguments, enrich the experimental evaluation of our algorithm and solidify our final conclusions, based on which we have carefully prepared a revised manuscript with the corresponding changes highlighted in blue. Here, we briefly summarize our major changes in the revised manuscript, as follows.

- **1) Further explanation of our FedSMU algorithm.**
  We have provided additional clarifications regarding the influence of stale momentum caused by partial participation in Appendix B.1. The precise meaning of the transmitted 1-bit has been explained in Appendix B.2, and the negative effects of extending our algorithm to multi-bit quantization have been discussed in Appendix B.3.

- **2) Further explanation of our theoretical analysis.**
  We have supplemented the motivation behind the assumptions made in our work and compared them with those in FedLion, as detailed in Appendices C and E.2. Additionally, the impact of the client participation rate and the number of local updates on the convergence rate have been analyzed in Appendix E.3. Furthermore, omitted detailed proofs and analyses have been incorporated into Appendices D and E.1.

- **3) Additional experimental results of FedSMU.**
  We have conducted experiments to evaluate the impact of model dimension on the performance of FedSMU in Appendix E.4. Additionally, we have provided the convergence curve of the algorithm, illustrating the relationship between communication rounds and performance, in Appendix F.1. The wall-clock time required for the experiments has been presented in Appendix F.2. We have also introduced a new algorithm in Appendix G.6 to validate the correlation between magnitude uniformity, data heterogeneity, and accuracy. Furthermore, we have extended the comparison to a new dataset, MNIST, in Appendix G.2 and re-evaluated the algorithm's performance using an alternative definition of generalization in Section 5.2.4.

- **4) More comparison with other methods.**
  We have extensively discussed and compared several related algorithms, including D-Lion, EF21, FedAMS, and FedCAMS in Appendix G.3 to Appendix G.5.

- **5) Further details revised and clarified.**
  We have tried our best to proofread the manuscript and improve the clarity of the text in the revised version, including rectifying typos, adjusting figure 1 scaling, adjusting the paper structure (by moving Fed-LocalLion and Fed-GlobalLion to the appendix G.1), and incorporating more relevant references, such as MARINA, EF21, and DIANA, into the Related Work section.

---

### Meta-Review · Area_Chair_EGrA · 2024-12-23

**Metareview:**

The paper introduces FedSMU, a federated variant of the Lion optimizer, designed to address communication efficiency in federated learning (FL). The FedSMU algorithm uses the Lion optimizer's sign-based updates, reducing client-to-server communication costs to 1-bit per parameter. The paper also provides a convergence analysis for both FedSMU and the original Lion optimizer, and compares to other compression-based methods.

Selected strengths:
- compared to FedLion, the design of FedSMU is better suited for FL due to lower communication cost
- in experiments, FedSMU performs better than other baselines the authors chose to compare to
- the paper introduces an interesting metric (magnitude uniformity) as a measure of heterogeneity in training

Selected weaknesses:
- missing important closely related work
- strong assumptions compared to other existing methods
- some answers to the reviews were not satisfactory, and showed relatively weak understanding of the frontiers of the FL literature
- a large number of imprecise & confusing statements in the paper text

I believe the paper needs a major revision, after which it will be in a much better shape. I encourage the authors to thoroughly revise the work in the light of the most relevant criticism raised by the reviewers.

**Additional Comments On Reviewer Discussion:**

While many questions and minor issues were handled in the rebuttal and ensuing discussion, the reviewers were not convinced that the paper, even after these changes, was strong enough to be accepted. 3 our of 4 reviewers still leaned towards rejection.

---

### Decision · Program_Chairs · 2025-01-22

Reject